https://doi.org/10.1038/s41467-020-15226-8　　**OPEN**

# Small molecule regulated sgRNAs enable control of genome editing in *E. coli* by Cas9

Roman S. Iwasaki[1], Bagdeser A. Ozdilek[1,3], Andrew D. Garst[2,4], Alaksh Choudhury[2,5] & Robert T. Batey [1✉]

CRISPR-Cas9 has led to great advances in gene editing for a broad spectrum of applications. To further the utility of Cas9 there have been efforts to achieve temporal control over its nuclease activity. While different approaches have focused on regulation of CRISPR interference or editing in mammalian cells, none of the reported methods enable control of the nuclease activity in bacteria. Here, we develop RNA linkers to combine theophylline- and 3-methylxanthine (3MX)-binding aptamers with the sgRNA, enabling small molecule-dependent editing in *Escherichia coli*. These activatable guide RNAs enable temporal and post-transcriptional control of in vivo gene editing. Further, they reduce the death of host cells caused by cuts in the genome, a major limitation of CRISPR-mediated bacterial recombineering.

[1] Department of Biochemistry, University of Colorado at Boulder, Boulder, CO 80309, USA. [2] Department of Chemical and Biological Engineering, University of Colorado at Boulder, Boulder, CO 80309, USA. [3] Present address: Department of Genetics, University of Georgia, Athens, GA, USA. [4] Present address: Inscripta Inc., Boulder, CO, USA. [5] Present address: IAME, Inserm U1137, Faculté de Médecine Université de Paris, Site Xavier Bichat, 16 rue Henri Huchard, Paris 75018, France. ✉email: robert.batey@colorado.edu

The utility of CRISPR for genome editing in *E. coli* has been demonstrated in various studies[1–4]. One of those methods, called CRISPR-enabled trackable genome editing (CRE-ATE)[5], uses plasmid-based recombineering, as opposed to the use of oligonucleotides, enabling easy tracking of the mutations in a library. However, CREATE, like other CRISPR-based editing technologies, suffers from low transformation efficiencies[1,5,6] caused by the lethality of dsDNA breaks in bacteria and from associated issues, such as biases in multiplexed libraries towards non-cutting sgRNAs. For this reason, we sought to develop inducible sgRNAs that allow timing and titration of the nuclease activity to alleviate these issues. Inducible CRISPR systems have previously been developed but are designed for CRISPRi applications and/or use in mammalian cells[7–19]. Due to the inherent leakiness of these systems they cannot be applied to inducible gene editing in bacteria—the same reason why inducible promoters could not be used to solve the problem[5,20]. Therefore, we show how to use the CREATE technology as a platform to develop ligand-switchable sgRNAs to control the initiation of gene editing.

## Results

**Selection of switchable aptamer-gRNAs from a library**. We used the CREATE technology as a platform to develop ligand-switchable sgRNAs to control the initiation of gene editing. As a first step, we replaced the tetraloop used to fuse the crRNA and tracrRNA[21] with an in vitro selected small molecule binding aptamer (Fig. 1a). This site is highly tolerant of insertions[22] and a theophylline aptamer-sgRNA fusion with an unchanged 2 × 4 internal loop (IL) is constitutively active but when the IL nucleotides were all substituted with uridine, gene editing activity was completely abolished (Supplementary Fig. 1). This confirms that nucleotides in the IL are critical for nuclease activity[23] via base specific interactions with the REC1 domain of Cas9[24]. We hypothesized that alternative sequences could communicate ligand-dependent conformational changes in the aptamer to Cas9 to regulate nuclease activity (Fig. 1b). To generate switchable aptamer-sgRNAs (agRNAs), we randomized a region including the IL and a small helix (upper stem) which connects the aptamer with the sgRNA.

A fully randomized 14 nucleotide (14N) library was subjected to an in vivo survival assay in *E. coli* (Supplementary Discussion 1). Each agRNA library was cloned into a plasmid vector that constitutively expressed the agRNA and contained the template for homologous repair of the cut site targeted by the agRNA as described in the CREATE[5] protocol. The cloned plasmid library was transformed into *E. coli* MG1655 to enrich theophylline-dependent agRNAs using a *galK* selection assay[25] in liquid culture. The MG1655 strain used for the selection also carries the pSIM5[26] plasmid which expresses the λ-red proteins from a heat-inducible promoter and pX2-Cas9[27] which expresses Cas9 from the arabinose-inducible pBAD promoter. This strain will be referred to as MGλ9.

In the first selection step MGλ9 was transformed with the agRNA library in the absence of theophylline. In this step, switchable agRNAs in the library are inactive. Expression of Cas9, but not the λ-red proteins was induced so constitutively active agRNAs that target the *galK* gene generated a double-stranded DNA break leading to cell death. Therefore, constitutively active agRNA constructs were eliminated in this selection step (Fig. 1c).

The counterselection step was facilitated by the Cas9-mediated recombineering technology CREATE[5]. First, the MGλ9 strain was transformed with the recovered plasmid library from the first selection step in presence of theophylline, inducing switchable agRNAs to cut the *galK* gene. Heat-induced expression of λ-red

proteins enabled repair of the DNA. Homologous recombination using a template provided on the agRNA plasmid introduced a premature stop codon into the *galK* gene preventing the host bacteria from fermenting galactose. We could then select for edited cells by growing in minimal M63 media containing the toxic galactose analog 2-deoxygalactose; only recombined bacteria that don't metabolize 2-deoxygalactose survive. Therefore, bacteria expressing activatable agRNAs in the presence of theophylline are enriched in this step (Fig. 1c). We iterated these selection/counterselection steps three times.

Candidates from this enriched library were analyzed to inform an improved library design. After three selection/counterselection steps, cells were plated on M63 selection medium[25] and 150 colonies were picked. We transformed each isolated candidate into unedited *E. coli* to screen for theophylline-responsive constructs using a red/white colony assay[28] to quantify the percentage of edited bacteria. Colonies with an unedited *galK* site will appear red because acidification of the media through fermentation of galactose causes the dye to turn red. Edited colonies appear white, because they cannot ferment galactose (Supplementary Fig. 2). The percentage of white colonies is considered the editing efficiency. Gene editing was induced as previously described with or without theophylline for 3 h after which the cells were plated on the MacConkey agar and the editing efficiencies were compared.

**Screening yields protospacer-independent agRNAs**. Analysis of the 150 colonies from the 14N theophylline agRNA library after the selection yielded 16 theophylline-responsive constructs (Supplementary Table 1), of which 3 constructs (A1, A9, A14) were found twice. In these sequences we observed a strong preference for Watson-Crick base pairs in the regions flanking the IL (Supplementary Table 2). Based upon this insight we generated more restricted libraries in which three base pairs were fixed, which we predicted would contain a higher percentage of switchable agRNAs within a lower total number of sequences. Three 8N libraries (~$6.6 \times 10^4$ sequences) were created that included two G-C pairs in the upper stem and one base pair at the 3′-side of the repeat/antirepeat helix (Fig. 1b). Both libraries were transformed in sufficient numbers so as to ensure complete library coverage with 95% confidence[29]. After the *galK* selection assay was used to enrich switchable constructs from those libraries, 20 colonies were screened from each library. Three sequences (GC7, GC10, GU10) out of the 60 from the three libraries were found twice. The low redundancy among the sequences of the initial and optimized libraries shows that even after enrichment our screening only covered a fraction of the switchable constructs. The editing efficiencies of the individual constructs from these libraries are displayed in Supplementary Table 1. Although most screened agRNA constructs were constitutively inactive (Fig. 1d), 10 out of 60 constructs from the restricted theophylline libraries showed a combined >40% editing efficiency and >10-fold increase in editing efficiency when induced (Fig. 1d) as opposed to 7 out of 150 from the initial screen. This result indicates that the optimization of the library design was successful in increasing the fraction of switchable constructs in the library.

The 17 promising agRNA constructs were selected for further analysis. The *xylA* gene was targeted with these 17 agRNAs to test whether a different spacer sequence affects the performance. Editing was quantified with the same red/white screen. Two agRNAs (A9 and GU19) showed an editing efficiency at *xylA* similar to the *galK*1 site used during selection, while 9 others showed >10% editing at *xylA* (Supplementary Fig. 3a). A9, GU19 were tested at further genomic *galK* sites, together with A1 as a

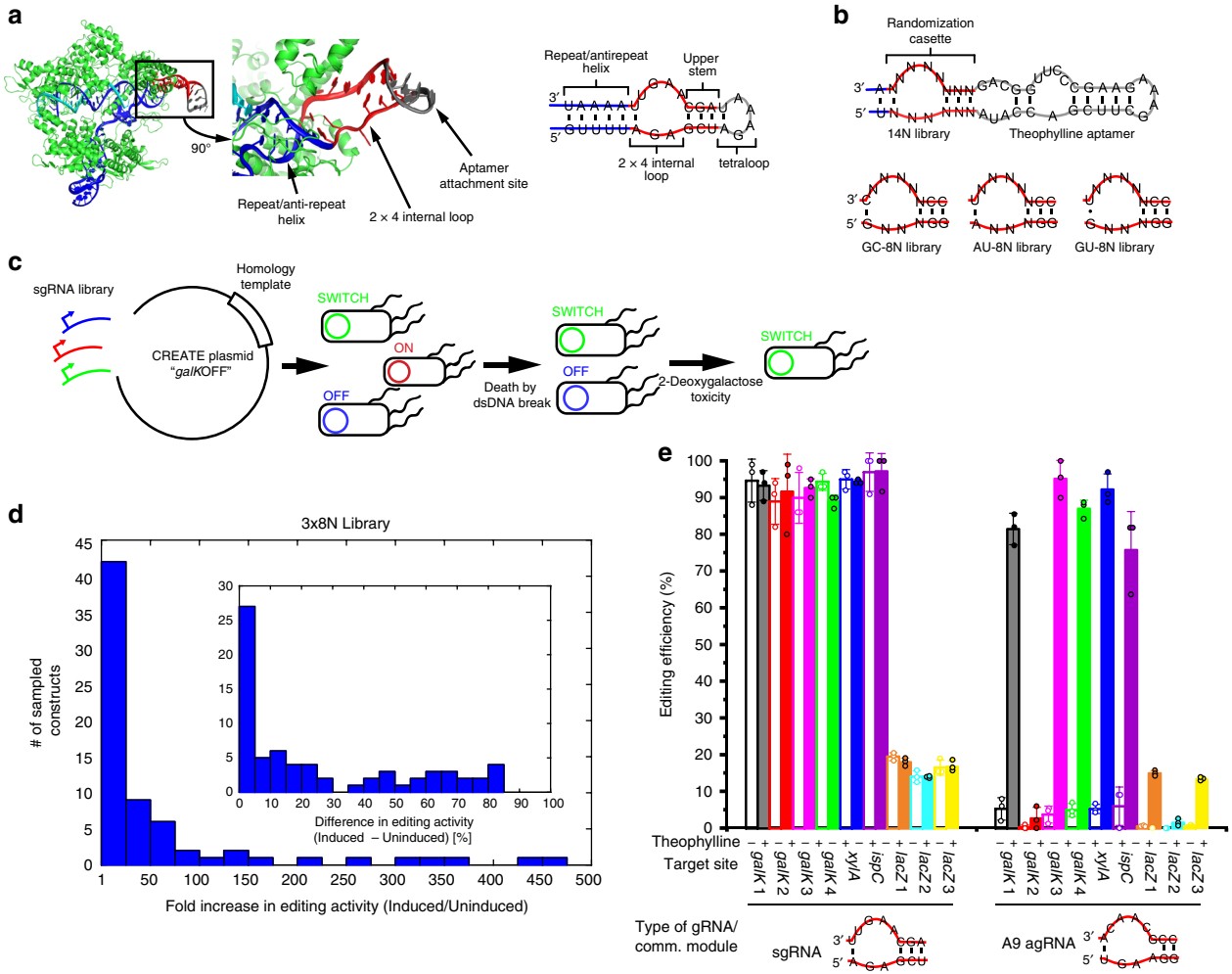

**Fig. 1 Design, selection and screening of agRNA libraries. a** The theophylline aptamer is inserted into the sgRNA at the site of the tetraloop (gray) used to fuse the guide and tracrRNAs (blue and red). **b** The 2 × 4 internal loop and flanking helices was randomized to yield agRNA libraries. **c** Switchable constructs are enriched in the two steps of the *galK* survival selection. **d** Overview of constructs that were sampled from the enriched plasmid library after the *galK* survival selection. Editing was induced with 1 mM theophylline. Left panel, agRNAs are binned according to the difference in editing efficiency in an induced and uninduced population. Inset panel, agRNAs are binned according to the fold increase in editing efficiency upon induction. **e** Inducible gene editing was tested at nine different genomic sites. $n = 3$ biological replicates. (Full circles: 1 mM theophylline; empty circles: no theophylline. Bars indicate average value; error bars indicate standard deviation.) Source data are provided as a Source Data file.

control. Also, it was confirmed by Sanger sequencing that the phenotypic change in the MacConkey assay corresponded to a designed genomic edit (Supplementary Fig. 4). When targeting three more alternative sites at the *galK* gene, A9 outperformed GU19 and was consequently tested at additional genomic sites in the *lacZ* and *ispC*[30,31] genes (Fig. 1e, Supplementary Fig. 3b). Editing at the *lacZ* gene was quantified with an X-gal screen and editing at the *ispC* gene was quantified with Sanger sequencing. Two of the *lacZ*-targeting agRNAs and the *ispC*-targeting agRNA exhibited editing efficiencies similar to unchanged sgRNAs. In total, out of nine sites, seven were targetable with the agRNA A9. This confirms that agRNA A9 is sufficiently re-targetable so that only testing a few sites will most probably reveal a high-efficiency target site. This demonstrates that using the agRNA A9 does not impose an additional workload or delay because even when using wt sgRNAs it is common to test a few target sites before a site is identified that can be edited efficiently. The observation that several agRNAs only act at the selected *galK*1 site highlights that the CRISPR-Cas9 system can be selected to act at a single sequence, a feature that might be exploited for reducing off-target effects.

**Binding of ligand to aptamer regulates endonuclease activity.** To investigate the effect of ligand binding on the gene editing, the concentration and induction time with theophylline was systematically varied. This showed that the editing efficiency increased with the concentration of theophylline and induction time (Fig. 2a, Supplementary Fig. 5). Even though no induction times longer than 3 h were tested, the induction could be prolonged if desired because theophylline is non-toxic to the cells at these concentrations. The concentration of theophylline in the media needed for rapid editing greatly exceeds the $K_D$ of the isolated aptamer (400 nM[32]), an observation typical for synthetic or natural ligand-activatable RNA devices[33,34]. To test whether the binding of the ligand to the aptamer is the trigger of the agRNA activation, we introduced the C22A point mutation at the ligand-binding site of the theophylline aptamer domain[35]. As expected, this turned the theophylline aptamer into a 3-methylxanthine (3MX) aptamer, that was not activatable by theophylline (Fig. 2b, Supplementary Fig. 6). This not only illustrates that ligand recognition by the aptamer is necessary for agRNA activation, but also expands our toolbox by a different ligand-activatable agRNA.

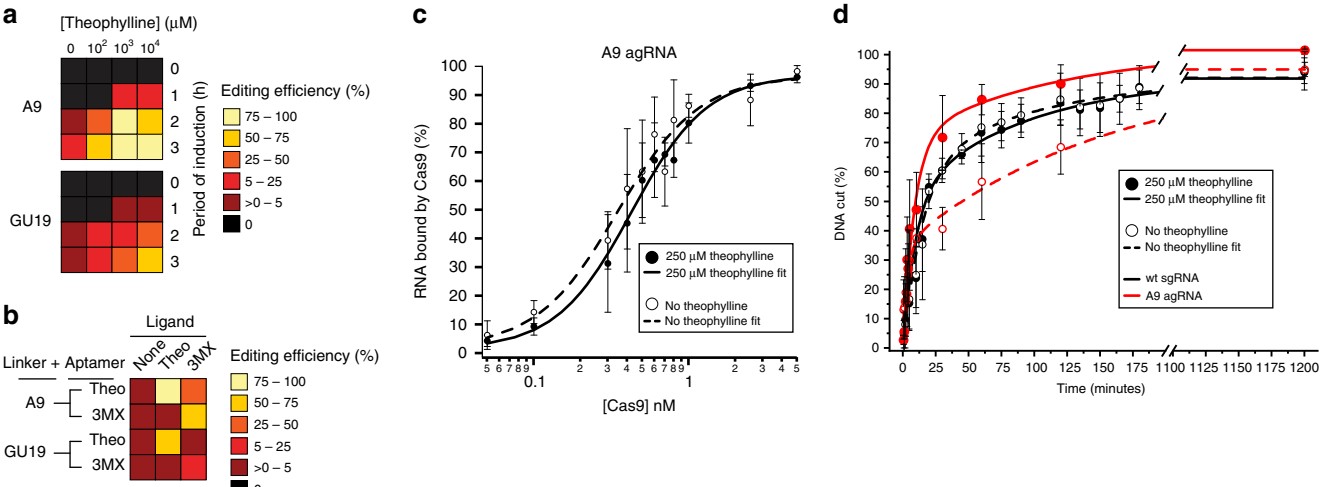

**Fig. 2 Characterization of selected agRNAs. a** Heat map of the editing efficiency at the *galK*1 site with the agRNAs A9 and GU19 as a function of the concentration of theophylline and the time of recovery with theophylline after transformation. In panels a and b, the color-coded values represent the average of three biological replicates. **b** Editing efficiency at the *galK*1 site was measured for agRNAs composed of the theophylline or 3MX- aptamer coupled via the linker A9 or GU19 in dependence of 1 mM theophylline (Theo) or 1 mM 3MX. Recovery time was 3 h. **c** Binding of Cas9 to sgRNA or agRNAs was observed and quantified via EMSA. Two representative binding curves for the A9 agRNA with and without 250 µM theophylline are displayed. **d** Kinetics of in vitro endonuclease activity. Cas9 was incubated with different sgRNAs and a dsDNA target with or without theophylline. Error bars indicate standard deviation from the mean, each circle represents an average of three measurements. Source data are provided as a Source Data file.

| Table 1 Dissociation constants of Cas9 and different gRNAs. | | |
| --- | --- | --- |
| **Type of gRNA** | **No theophylline** $K_{D,app}$ **(nM)** | **250 µM theophylline** $K_{D,app}$ **(nM)** |
| wt sgRNA | 0.50 ± 0.05 | 0.53 ± 0.05 |
| wt agRNA | 0.50 ± 0.03 | 0.48 ± 0.04 |
| A9 agRNA | 0.43 ± 0.03 | 0.35 ± 0.04 |
| GU19 agRNA | 0.48 ± 0.04 | 0.42 ± 0.02 |
| A1 agRNA | 0.45 ± 0.04 | 0.47 ± 0.05 |

To understand the effects of ligand binding to the aptamer on the activity of Cas9 we investigated binding of the agRNA to the Cas9 protein and cleavage of dsDNA target by the resultant RNP. Binding of $^{32}$P-labeled agRNA to Cas9 was quantified using an electrophoretic mobility shift assay (EMSA) (Supplementary Fig. 7) in the presence or absence of 250 µM theophylline. A1, A9 and GU19 agRNAs all exhibit the same apparent binding affinity to the Cas9 protein as the wild type sgRNA, independent of theophylline (Fig. 2c, Table 1). This indicates that despite the expected disruption of protein-RNA contacts in the randomized region, the RNP assembly is not ligand-dependent. Instead, we hypothesize that the ligand regulates the recognition and cleavage of the DNA target. To measure the endonuclease activity of the Cas9-gRNA complexes in vitro, Cas9 protein was assembled with wt sgRNA, A9 agRNA, GU19 agRNA or A1 agRNA with and without theophylline. The RNP complexes were then incubated with $^{32}$P-labeled dsDNA targets (Fig. 2d, Supplementary Fig. 8, 9). Although theophylline increases the nuclease activity of the RNP complex, the in vitro conditions greatly reduce the dynamic range of induction with theophylline. This inconsistency between functionality in vivo and in vitro might be caused by differences between the cellular environment and the conditions in vitro, illustrating the importance of using cell-based selections when developing cell-based tools.

**agRNAs can increase transformation and editing efficiency.** Equipped with temporally controllable agRNAs, we optimized the

gene editing technology CREATE[5]. One major drawback of CRISPR-mediated bacterial gene editing technologies, is cell death caused by dsDNA cuts[5], despite overexpression of λ-red proteins to increase the frequency of homologous repair. We hypothesize that the stress of transformation combined with rapid generation of dsDNA breaks synergistically leads to a high level of cell death and that temporally separating the two events can improve survival (Fig. 3a). Targeting the *galK1* site with the A9 or GU19 agRNAs resulted in a remarkable $10^4$-fold increase in number of transformants while maintaining ~80% editing efficiencies (Fig. 3b). Although a low transformation efficiency is not a significant issue when creating a single edit with a defined sgRNA, it dramatically impacts the ability to transform a library of sgRNA plasmids to generate a population of bacteria with different edits.

A critical issue faced in the preparation of libraries of sgRNAs for genome editing is that a portion of sgRNAs from the library is expected to be non-functional due to errors in oligonucleotide synthesis, sgRNA misfolding, or inefficient targeting. Transformants with non-functional sgRNAs do not suffer Cas9-mediated DNA cuts and have a strong fitness advantage over transformants with functional sgRNAs, resulting in a population dominated by wild type cells. To simulate the transformation of a library containing non-functional sgRNAs, bacteria were transformed with plasmid mixtures which contained four different plasmids targeting a different site each on the *galK* gene and varying amounts of non-targeting plasmid. Deep sequencing of the *galK* gene after editing revealed low (1–10%) editing efficiencies for both, the sgRNA and agRNAs and the efficiencies further decreased with increasing amounts of non-targeting gRNAs (Supplementary Fig. 10). Similarly, when only a single site on the *galK* gene (*galK*1) was targeted with a sgRNA, editing efficiencies dropped from 89 to 14% when 10% of the plasmids expressed non-targeting sgRNAs (Fig. 3c). It was sequence-verified that most of the unedited cells carried plasmids expressing non-targeting sgRNAs (Supplementary Table 3). This corresponds to an 8.6 ± 0.1-fold enrichment of the non-targeting plasmids. In contrast, using agRNAs A9 and GU19, the percentage of edited cells was maintained at 81% and 63%, respectively, which shows

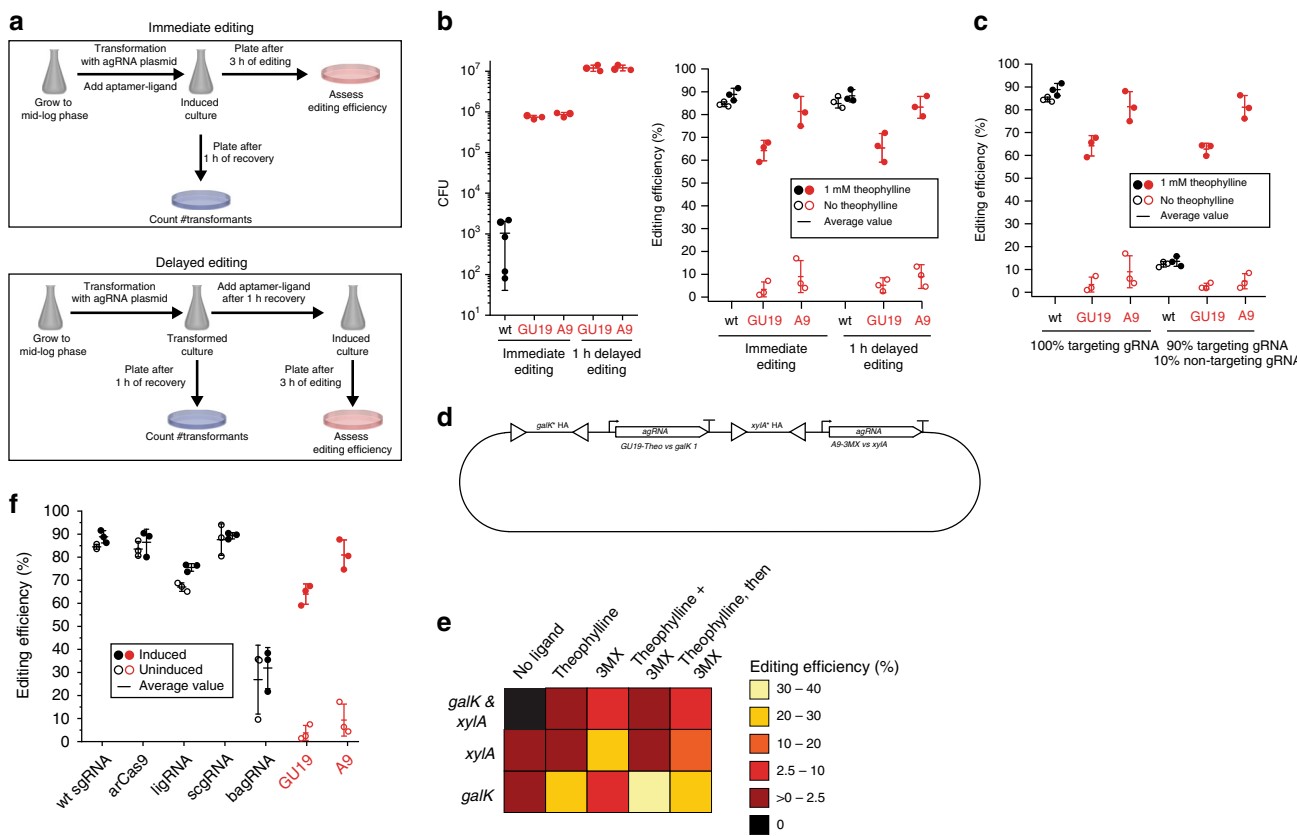

**Fig. 3 Aptamer-gRNAs enable high-throughput genome editing. a** Comparison of workflows with and without aptamer-regulated control of editing time point. **b** Transformation efficiencies of the MGλ9 strain with wt sgRNA and agRNAs that are induced immediately after transformation or 1 h after transformation. CFU are colony forming units 100 ng$^{-1}$ DNA and per 4 mL of recovery media. Error bars indicate standard deviation from the mean. $n = 3$–5 biological replicates. **c** Editing efficiencies with wt sgRNA and ligand-activatable agRNAs that are activated with theophylline 1 h after transformation. Targeting sgRNA refers to plasmids that express sgRNAs that target the *galK*1 site, non-targeting sgRNA refers to plasmids expressing a sgRNA that targets a site absent from the *E. coli* genome. Each dot represents a biological replicate and error bars indicate standard deviation from the mean. $n = 3$ biological replicates. **d** Plasmid design to enable multiplexed gene editing. The individual homology arms and agRNAs are identical to the ones previously described for the single-edit experiments. **e** Editing efficiency at the two genes *galK* (site 1) and *xylA* in dependence on agRNA induction with either 1 mM theophylline and/or 1 mM 3MX. The color-coded values represent the average of three biological replicates. **f** Editing efficiencies at the *galK* 1 site with various systems. wt sgRNA: unmodified single-guide RNA. arCas9[7]: an allosterically regulated Cas9 protein that is inducible with 0.1 mM 4-hydroxytryptophan. ligRNA[8]: a ligand-inducible sgRNA that is functionalized with a theophylline aptamer. scgRNA[9]: a signal-conducting sgRNA that was functionalized with a theophylline aptamer. bagRNA[11]: a blocked aptamer-gRNA that uses a theophylline-ribozyme to unblock the sgRNA spacer. ● Full circles: induced with 1 mM theophylline, unless stated otherwise. ○ Empty circles: no induction. Every circle represents one biological replicate. Error bars indicate standard deviation from the mean. $n = 3$ biological replicates. Source data are provided as a Source Data file.

that non-targeting plasmids were not enriched in the process. This result indicates that separating the stresses caused by transformation and double-stranded breaks can dramatically increase cell survival, possibly by decreasing the fitness differences between cells harboring functional and non-functional sgRNAs.

To ascertain whether agRNAs can facilitate creation of libraries of variants, the percentage of non-functional sgRNAs in the plasmid mix was varied from 0 to 50% in combination with four different plasmids, each expressing an agRNA targeting a different site on the *galK* gene. After induction of editing, cells were plated and the edits in the *galK* gene were analyzed via next-generation sequencing. In the absence of non-functional sgRNAs, editing efficiencies with the agRNAs were as low as with the sgRNAs, around 1–10% (Supplementary Fig. 10). Increasing amounts of non-functional sgRNAs further reduced the editing efficiencies. We cannot explain the drastic loss of editing efficiency in light of the single agRNA experiment (*vide supra*). However, we suspect that cells might be transformed with more than one plasmid such that the sgRNAs would target more than one site. This would lead to multiple genomic DNA breaks which

places a large burden on the cells so that any potential "escapees" can take over the bacterial population. *E. coli* does not typically survive two simultaneous dsDNA breaks, even with induction of the λ-red proteins to facilitate homologous repair[5].

**agRNAs enable introduction of combinatorial gene edits**. The issue of cell death caused by dsDNA breaks also limits the creation of combinatorial gene edits as only one mutation can be introduced in one round of CREATE editing, otherwise none of the transformants survive. We envisioned that our agRNAs would allow precise control over the timing of each editing event, facilitating sequential introduction of multiple designed edits into the same cell. We designed a plasmid that expresses a theophylline-regulated GU19-agRNA targeting *galK*1 and a 3MX-regulated A9-agRNA targeting the *xylA* gene. Furthermore, the plasmid also contains the homology arms necessary for repair and mutation of the respective cut sites (Fig. 3d). The plasmid construct was then used for a 3-h induction with either 1 mM theophylline, 1 mM 3MX or a combination of the two and the

transformed bacteria were plated on MacConkey agar that contained either galactose, xylose or both to assay editing of the galK and xylA genes with the red/white screen (Fig. 3e, Supplementary Fig. 11). When targeting both genes with sgRNAs, no transformants were observed because E. coli rarely survives two simultaneous dsDNA breaks. Therefore, no editing efficiencies could be calculated for the sgRNAs. As expected, adding theophylline to bacteria transformed with the plasmid that expresses two agRNAs induced editing of the galK gene but not the xylA gene and adding 3MX induced mostly editing of xylA and to a lesser extent galK. This corresponds to previously described promiscuity of the 3MX aptamer[35]. Interestingly, a small number of transformants carried both mutations as a result. Simultaneously adding theophylline and 3MX led mostly to galK editing and few xylA edits. Adding first theophylline and after 3 h adding 3MX for 2 h led to an improved editing efficiency of xylA and resulted in transformants carrying both genomic edits. Thus, agRNAs can achieve CRISPR-mediated multiple genome editing in bacteria in a single step on a scale that allows the coverage of commonly used libraries.

## Discussion

In summary, we have developed an in vivo selection approach that can be used to enrich ligand-activatable sgRNAs. By stringently selecting for switchable agRNAs with low background activity, we obtained agRNAs that overcome the issue of leakiness associated with other inducible systems and therefore enable regulated gene editing in E. coli. The agRNAs also allowed a drastic improvement in the throughput of CRISPR-mediated recombineering by increasing the transformation efficiency $10^4$-fold and reducing the bias for non-functional sgRNAs. Furthermore, combination of different guide sequences with aptamers that bind distinct ligands allows for multiplexing of the agRNA regulation enabling multiple, targeted mutations in a single experiment. All other inducible CRISPR systems that we used to edit the E. coli genome at the galK site were constitutively active (Fig. 3f). This is probably due to the fact that those systems were tested and developed for CRISPRi applications[36] or applications in mammalian cells[7,9,11]. Similarly, we anticipate that our agRNAs are not universally applicable, which reinforces the need to develop CRISPR tools that are tailored to various gene editing applications. With this study, we have shown a way forward in developing tools for regulatable gene editing in E. coli and other bacteria important in biotechnology.

## Methods

**Strains and plasmids.** For molecular cloning, the E. coli strain E. cloni® was used.

For the recombineering experiments, an E. coli MG1655 strain was used that contained two plasmids: The plasmid pSIM5, which expresses the λ-red proteins from a heat shock-inducible promoter and the X2-Cas9 plasmid (Addgene ID: 85811) that encodes Cas9 under control of the arabinose-inducible pBAD promoter. This strain will be referred to as MGλ9.

The sgRNA constructs were constitutively expressed from a CREATE vector that was based on the pUC19 backbone. A CREATE vector also contains a template to introduce mutations at the genomic cut site during homologous repair[5].

**Molecular cloning.** Homology-based cloning was used for construction of new plasmids. For generating plasmid backbones and inserts that contained homologous ends, PCR with the Q5® High-Fidelity 2X Master Mix was used. The annealing temperatures typically ranged from 60 to 72 °C. For the cloning of single inserts, CPEC[36] was used. Libraries were constructed via Gibson Assembly[37] using the NEBuilder® HiFi DNA Assembly Master Mix.

**Library design and preparation.** The double-stranded insert, containing the sgRNA constructs, was generated by PCR assembly of two single-stranded oligos. A linearized vector backbone containing homology arms was produced by PCR from the galKOFF plasmid using the primers theo-gRNA_BBF and theo-gRNA_BBR. Insert and backbone were assembled by Gibson Assembly. The product was gel purified using a QIAquick Gel Extraction Kit and transformed into E. cloni® bacteria via electroporation. The cells were recovered in SOB media for 1 h and

aliquots were plated on LB agar containing 100 μg/mL carbenicillin, which indicated that about $10^7$ CFU were recovered. The recovery culture was transferred to LB media containing 100 μg/mL carbenicillin, grown for ~14 h and then 150 μg plasmid library was harvested using a QIAprep Spin Miniprep Kit. Then, 0.5 μg of the plasmid library was electroporated into the MGλ9 strain 24 times to transform 12 μg DNA in total. The bacteria were recovered immediately in LB medium and aliquots were plated on LB agar after 1 h to estimate the number of CFUs. Then, the MGλ9 culture was subjected to the galK selection assay. The 24 transformations yielded a total MGλ9 CFU of $3 \times 10^6$.

**GalK selection assay.** We mostly followed protocols from Warming et al.[25], with the noticeable difference that the recombineering step was mediated by plasmids, using the CREATE technology, instead of using single-stranded oligos.

Negative selection: The library was transformed into MGλ9 cells without heatshock or addition of sgRNA-ligand and recovered at 30 °C for 5 h in LB containing 0.2% arabinose. The antibiotics chloramphenicol, kanamycin and carbenicillin were added 3 h after transformation. Constitutively active sgRNA constructs that allowed Cas9 activity in the absence of theophylline cause a double-stranded break, which leads to cell death.

Positive selection: The E. coli culture underwent recombineering as described in the section "CREATE Recombineering". However, after recovery the culture was not plated on MacConkey, but washed two times in M9 media to remove metabolizable sources of carbon and an aliquot was transferred to M63 selection media, which contained 0.2% glycerol, 0.2% 2-deoxygalactose, 1 mM MgSO$_4$ and kanamycin and carbenicillin, apart from the M63 salts. This media only permits growth of cells that introduced a stop codon into the galK gene. MG1655 cells typically plateau at a cell density of about OD$_{600}$ = 2 after 2–3 days in M63 media, so bacteria were added to an initial density of OD$_{600}$ < 0.2 to allow for enrichment of the editing cells via outgrowth. The cells were grown in the selection media at 37 °C for about 2–3 days or until an OD$_{600}$ of 1.5–2.0 was reached. Then, plasmids of the bacterial culture were harvested and the selection cycle can be repeated for further enrichment. After each transformation step, dilutions of the transformation culture were plated on LB agar to estimate the number of transformants.

Selection conditions were made increasingly stringent with increasing number of cycles by lowering the concentration of sgRNA ligand progressively from 1 mM to 250 μM theophylline and by progressively shortening the time available for editing before transfer to selection media from 5 to 1 h. In the third selection cycle, the positive selection was not carried out in liquid culture. Instead, the washed E. coli culture was plated on agar plates containing the M63 selection media. After incubating the plates at 37 °C for 2–3 days, colonies were picked from the M63 plates for screening.

**CREATE recombineering.** Original protocol: We mainly followed the protocol established by Garst et al.[5]. LB containing kanamycin and chloramphenicol was inoculated with MGλ9 cells and grown overnight at 30 °C. The stationary culture was diluted 100 fold in LB, containing chloramphenicol, kanamycin and 0.2% arabinose and grown to an OD$_{600}$ = 0.4–0.6 in 25 mL LB. The culture was heat-shocked in a shaking waterbath at 42 °C for 15 min to induce expression of the lambda-red proteins from pSIM5. Then, the culture was washed in two volumes of ice-cold deionized water, resuspended in 500 μl deionized water and 50 μl of the cell suspension was electroporated with 200 ng of a CREATE plasmid that encodes sgRNA and homology template. The transformants were recovered in 4 mL of LB, containing 0.2% arabinose and optionally 1 mM theophylline and shaken at 37 °C for 3 h before plating on MacConkey Agar. To estimate the number of colony forming units (cfu), aliquots were plated on LB agar 1 h after transformation.

Aptamer-protocol: LB containing kanamycin and chloramphenicol was inoculated with MGλ9 cells and grown overnight at 30 °C. The stationary culture was diluted 100 fold in LB, containing chloramphenicol and kanamycin and grown to an OD$_{600}$ = 0.4–0.6 in 25 mL LB. Then, the culture was washed in two volumes of ice-cold deionized water, resuspended in 500 μl deionized water and 50 μl of the cell suspension was electroporated with 200 ng of a CREATE plasmid that encodes sgRNA and homology template. The transformants were recovered in 4 mL of LB, containing 0.2% arabinose and shaken at 30 °C for 1 h. The culture was heat-shocked in a shaking waterbath at 42 °C for 15 min and 1 mM theophylline was added to the culture. The culture was shaken at 37 °C for 3 h before plating on MacConkey Agar. To estimate the number of cfu, aliquots were plated on LB agar 1 h after transformation.

Multiplex-protocol: For the purpose of first adding theophylline and later 3MX to independently induce two different agRNAs, the protocol was slightly modified. First, the bacteria were prepared and transformed as in the original protocol, 1 mM theophylline was added to the media and the culture was shaken at 30 °C for three hours. Then, the culture was heat-shocked again at 42 °C for 15 min to induce expression of the lambda-red proteins again and 1 mM 3MX was added to the media. The culture was shaken at 37 °C for 3 h and then plated.

**Red/white screening.** In order to assess the frequency of gene editing, a red/white screen was used to visualize the introduction of a stop codon into the galK gene by CREATE. The original CREATE protocol was used for inducing gene editing and after 3 h of recovery (with or without the agRNA ligand) a series of dilutions of the

cultures were plated on MacConkey Agar plates containing 1% galactose. Bacterial colonies with a mutated *galK** site appear white on the plates, whereas non-edited colonies appear red. The editing efficiency could be calculated by dividing the number of edited colonies over the total number of colonies. By comparing the editing efficiency in presence versus absence of the ligand, a dynamic range of the in vivo activity could be determined.

When screening the enriched libraries for switchable constructs, the transformants were recovered in 4 mL of LB broth, a dilution series was plated for every construct and the experiment was only carried out once due to the large volume of colonies screened (~250 in total). When the constructs A9, GU19 and A1 were characterized in detail, triplica of the experiment were carried out. This may account for differences in the observed editing efficiency from the initial screen (Supplementary Fig. 2c).

**Quantification of editing at the *lacZ* and *ispC* sites.** Editing at the *lacZ* target sites was visualized with an X-gal color screen and calculated the same way as with the MacConkey color screen. Editing at the *ispC* gene was quantified by Sanger sequencing of PCR amplicons from 11 to 12 bacteria colonies for each replica.

**Calculation of plasmid enrichment.** When the E. coli strain MGλ9 is transformed exclusively with the CREATE galKOFF plasmid that expresses wt sgRNA that targets the galK gene, 100% of the transformants carry the targeting plasmid and 88.9% of the bacterial population is edited after the CREATE procedure. When transforming a mixture of targeting plasmid and non-targeting plasmid in a 9:1 ratio, 80% of the bacteria are expected to be edited according to $0.9 \times 88.9\% = 80.1\%$, assuming no bias for the non-targeting plasmid. However, only 13.5% of the plated colonies were edited, which means that 15.2% of plated bacteria are expected to carry the targeting CREATE plasmid, when considering that only 88.9% of bacteria that are transformed with the targeting plasmid get edited: $\frac{13.5\%}{88.9\%} = 15.2\%$. This means, that $100 - 15.2 = 84.8\%$ of the plated bacteria carry the non-targeting plasmid, which is a ~ 8.5-fold enrichment over the 10% of non-targeting plasmid in the transformation mix.

A formalized description would be:

$P_{nt}$ = Fraction of plasmids with non-targeting sgRNA

$E_t$ = Editing efficiency of the targeting sgRNA

$E_{nt}$ = Editing efficiency obtained from mix with non-targeting plasmids as observed

$T_t$ = Fraction of transformants with the targeting sgRNA plasmid

$T_{nt}$ = Fraction of transformants with non-targeting sgRNA plasmid

$$\text{Enrichment of non} - \text{targeting sgRNA plasmid} = \frac{T_{nt}}{P_{nt}} = \frac{1 - T_t}{P_{nt}} = \frac{1 - \left(\frac{E_{nt}}{E_t}\right)}{P_{nt}}$$

The error is given as ± standard deviation.

**Next-generation sequencing.** One milliliter of 1× PBS was added to bacterial colonies on LB agar plates and colonies were scraped from the plate. Fifty microliters of the cell suspension were centrifuged and washed twice with PBS. Then, the cells were resuspended in 50 μL TE buffer (pH 8.0) and boiled at 95 °C for 10 minutes. Next, a region of the galK1 gene was PCR-amplified from 1 μl of sample with OneTaq polymerase in a final volume of 50 μL. The primers that were used contained Illumina Nextera adapters (Supplementary table 2). PCR was performed as follows: 94 °C for 10 min, 10 cycles of 94 °C for 30 s, 54 °C for 30 s, 68 °C for 60 s and one cycle at 72 °C for 5 min. Each sample was amplified with a unique Nextera barcode. Sequencing was carried out with the Nextera next generation sequencing kit.

**Cas9 expression and purification.** *Streptococcus pyogenes* Cas9 (pMJ915) construct was a gift from Jennifer Doudna (Addgene plasmid # 69090)[38]. The construct was transformed into BL21 (DE3) Rosetta *Escherichia Coli* cells. 10 mL LB-Ampicillin bacterial culture was grown overnight and then inoculated into 1 L LB medium. Culture was grown at 37 °C until $OD_{600nm}$ reached around 0.6. The culture was cooled down to ~20 °C in a cold water bath and protein expression was induced by adding 0.5 mM Isopropyl b-D-1-thiogalactopyronoside (IPTG). The culture was grown in a 20 °C shaker overnight. Bacterial cells were pelleted at 1,500 g and resuspended in lysis buffer (1 M KCl, 20 mM HEPES pH 7.5, 20% glycerol, 1 mM TCEP, 10 mM Imidazole). Cells were lysed using an Emulsiflex C3 homogenizer. The cell lysates were clarified by centrifugation at 17,000 × g for 30 min. Polyethyleneimine (PEI) was used to precipitate the nucleic acid contaminants[39]. The supernatant (35 mL) was put into a beaker 4 °C and 250 μL 5% PEI was slowly added during stirring. The supernatant was stirred for 15 more minutes. Then, it was centrifuged at 12,000 × g for 20 min to pellet the nucleic acid contaminants. The supernatant was taken and its PEI concentration was brought to 0.1% and stirred at 4 °C for 15 min. It was centrifuged at 12,000 × g for 20 min. Then, the supernatant was incubated with Ni-NTA sepharose beads on an orbital shaker for 1 h at 4 °C. Beads were centrifuged at 300 × g for 2 min and washed three times in lysis buffer and once in lysis buffer supplemented with 100 mM Imidazole. Proteins were eluted in lysis buffer supplemented with 250 mM Imidazole. The eluate was concentrated and the buffer was exchanged (20 mM HEPES pH 7.5, 500

mM KCl, 1 mM TCEP, 10% glycerol). Then, size exclusion purification was conducted on a Hiload 16/600 Superdex 200 column (AKTA Purifier system (GE Healthcare)). Cas9 protein was purified as monomer, based on comparison to size standards. Final protein concentration was calculated using molar extinction coefficient as determined using the Expasy-Protparam tool and the absorbance at 280 nm. Cas9 was kept at -20 °C.

**In vitro transcription and purification of sgRNAs.** DNA template for RNA transcription was amplified by using PCR and transcribed by T7 RNA polymerase[40]. For a 3 mL transcription reaction, 1.9 mL ddH$_2$O, 0.3 mL transcription buffer (10×), 100 μl MgCl$_2$ (1 M), 125 ul from each rNTPs (100 mM), 200 μl PCR template, 31 μl DTT (1 M), 25 μl inorganic pyrophosphatase (20 U/μl), 50 μl T7 RNA polymerase were assembled in a 15 mL canonical. The reaction was vortexed and incubated at 37 °C for 2 h. Then, 6 mL ethanol was added to the reaction and kept at −80 °C for at least 30 min or at 20 °C for overnight to precipitate the RNA. The tubes are centrifuged at 4000 × g and at 4 °C for 15 min. The supernatant was discarded and pellet was left for air-drying to evaporate the ethanol at room temperature. The pellet was suspended in 2 mL of 8 M urea, 500 μL 0.5 M EDTA pH 8.0, and 1 ml of formamide loading dye. To re-suspend all of the precipitate the tube was vortexed vigorously. To ensure the complete denaturation of the RNA, samples were heated at 65 °C for 5 min and vortexed vigorously until getting a clear solution. Transcripts were purified using denaturing polyacrylamide gel (6–10% 29:1 acrylamide/bisacrylamide, 1x TBE buffer (0.1 M Tris base, 0.8 M boric acid, 1 mM Na$_2$EDTA), 8 M urea). RNA bands were visualized by putting the gel on a fluorescence TLC plate and shadowing the RNA with short-wave UV in a dark room. Full-length transcripts were excised from the gel and the gel pieces were further crushed into small pieces inside a tube. 0.5x TE buffer was added to the tube and the mix was shaken gently at 4 °C for 2 h to extract the RNA. RNA from the supernatant was concentrated using centrifugal concentrators with a 10 kDa molecular weight cutoff (Amicon Ultra, 0.5 mL) and buffer (0.5× TE) exchange was performed by the same method. Final RNA concentration was calculated using the absorbance at 260 nm and the molar extinction coefficient as determined using an extinction coefficient calculator that calculates the extinction coefficients by summing of the individual extinction coefficients for each nucleotide in the RNA. The RNA was aliquoted into 5 μl volumes and stored at −20 °C until use.

**Body radiolabeling reaction of sgRNAs.** 100 μl in vitro RNA transcription reaction was prepared with an adenine ribonucleotide concentration that is 10-fold lower than the standard reaction concentration. 20 μCi ATP [α-32P] was added and the reaction was carried out with T7 RNA polymerase at 37 °C for 2 h. MicroSpin G25 columns were used to remove unincorporated nucleotides from the labeling reactions. Radiolabeled transcripts were purified using 6% denaturing polyacrylamide gel (29:1 acrylamide/bisacrylamide, 1× TBE buffer, 8 M urea). The gel was exposed using a phosphoimager for about 10–15 min and the screen was imaged by using a Typhoon PhosphoImager. The image was printed out with actual sizes. The gel was placed on top of the printed image and the corresponding RNA band was excised from the gel. Gel pieces were put into 2 mL eppendorf tubes and crushed into small pieces by using 1 mL pipette tip. 0.5× TE buffer with 0.3 M sodium acetate (pH 5.3) was added into the tube and left for elution by rotating at 4 °C for 2 h. The radiolabeled RNAs were precipitated with ethanol and glycogen at −80 °C for 30 min (or overnight at −20 °C) and centrifuged at 17,000 × g for 30 min at 4 °C. Precipitated RNA was resuspended in 0.5× TE buffer and quantified by liquid scintillation counting.

**5′-end radiolabeling of target DNA molecules.** Forward and reverse strands of target DNA molecule (Supplementary Table 2) were chemically synthesized (Eurofins). The 5′ end of one of these strands was radiolabeled as following: forward or reverse strand of DNA (1 μM final concentration), 10× T4 PNK buffer, T4 PNK enzyme (10 U/μL, 1 μl was added for 20 μl reaction volume), γ-³²P ATP (10 μCi/μl, 1 μl was added for 20 μl reaction volume), x μl ddH$_2$O to make the desired final volume. Reaction was incubated at 37 °C for 30–45 min. Unincorporated nucleotides were removed by MicroSpin G25 columns. PCR purification protocol was used to purify the labeled DNA strand. Radiolabeled DNA was eluted with appropriate amount of ddH$_2$O to achieve 1 μM final concentration. To get a double-stranded DNA molecule, radiolabeled and the complementary strands were annealed as follows: 1 μl forward strand (1 μM), 1 μl reverse strand (1 μM), 10 μL NaCl (1 M) and 88 μl 0.5 x TE buffer (10 mM Tris-HCl pH 8.0, 1 mM EDTA) are combined in a PCR tube. Then, the tubes are heated to 95 °C and cooled down to 16 °C at −1 °C/30 s. The dsDNA molecule was kept at −20 °C until use.

**Electrophoretic mobility shift assay (EMSA).** EMSA experiments were carried out to measure dissociation constants (Kd) of the Cas9 binding to sgRNAs (Supplementary Fig. 7) reactions. sgRNAs were radiolabeled as described above. They were heated at 95 °C for 3 min and snap cooled before addition to the binding reactions. Cas9 proteins (0–5 nM) were incubated with trace amount of (~0.05 nM) radiolabeled RNA molecules in binding buffer containing 20 mM HEPES pH 7.5, 200 mM KCl, 5 mM DTT, 5% glycerol, 0.01% NP40 with or without 250 μM theophylline. A native polyacrylamide (6%, 29:1 acrylamide/bisacrylamide) supplemented with 0.5x TB (45 mM Tris-HCl, 45 mM borate, pH 8.1) buffer was used to separate the bound and unbound sgRNA species. Gels were dried and

subsequently imaged using a Typhoon PhosphoImager (Molecular Dynamics) and the signals were quantified with the ImageQuant software suite and ImageJ 1.52a. Quantified data was fit to a standard two-state binding isotherm using Igor Pro 8 (Wavemetrics) and Origin 2018b, allowing calculation of both dissociation constants and Hill Coefficients.

**Endonuclease assay**. Cas9 (final concentration 200 nM) and sgRNA (final concentration 300 nM) were preincubated in the reaction buffer (50 mM HEPES pH 7.5, 200 mM NaCl, 5% glycerol, 2 mM DTT, 5 mM MgCl$_2$ with or without 250 µM theophylline) at room temperature for 20 min. Then, <0.4 nM 5′-end radiolabeled dsDNA was added to the reaction and incubated at 37 °C. Ten microliters of aliquots were taken at the indicated time points. The reactions were quenched with equal volume of quenching solution (8 M urea, 1× TBE, 0.01% bromophenol blue and 0.01% xylene cyanol) and boiled at 95 °C for 3 min. Cleavage products were resolved on 10% denaturing polyacrylamide gel. Gels were dried and subsequently visualized using the Typhoon PhosphoImager and the signals were quantified with ImageJ 1.52a. A two-term exponential model was fit to the data, using the program Origin 2018b, and time constants of the DNA cleavage were calculated from the fit.

$$\text{DNA cut} = \frac{A_1}{\left(e^{\frac{t}{t_1}}\right)} + \frac{A_2}{\left(e^{\frac{t}{t_2}}\right)}$$

In some cases, the fit did not converge, which is why in those cases no standard deviation for the constant is given.

**Reporting summary**. Further information on research design is available in the Nature Research Reporting Summary linked to this article.

## Data availability

The authors declare that the main data supporting the findings of this study are available within the article and its supplementary information files. The source data for Figs. 1d, e, 2–d, 3b, c, e, f and Supplementary Figs. 1, 3a, b, 5–8, 10, 11 are provided as a Source Data File. Raw Illumina sequencing data of DNA amplicons is deposited at the NCBI Sequence Read Archive under accession code SAMN14149857 (Bioproject: PRJNA607866).

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

## Acknowledgements

We gratefully acknowledge G Pines and MC Bassalo for insightful discussions. This work was supported by a grant from the National Institutes of Health (R01 GM073850) to R.T.B.

## Author contributions

R.S.I. and R.T.B. contributed to the concept and design of all experiments. A.D.G. contributed to design of the in vivo experiments. R.S.I. carried out all in vivo experiments. B.A.O. and R.S.I. designed and performed the in vitro experiments. A.C. carried out analysis of the deep sequencing data. R.S.I. and R.T.B. wrote the manuscript.

## Competing interests

The authors R.T.B., R.S.I., and A.D.G. declare the following competing financial interests: R.T.B., R.S.I., and A.D.G. have submitted a patent application (No. PCT/US2020/013718) related to the use of agRNA for gene editing. A.D.G. has competing financial interests in Inscripta Inc., a company that is commercializing multiplexed genome editing platforms. The authors B.A.O. and A.C. declare no competing interests.
