## [Peer Review File · Nature Communications]

Reviewers' Comments:

Reviewer #1:

Remarks to the Author:

In the submitted manuscript, the authors developed novel ligand-switchable sgRNAs for genome editing in bacteria. I like the concept of a screen to identify functional theophylline-responsive or 3MX-responsive sgRNAs. However, the topic of conditional CRISPR-Cas systems is very well researched over the past few years (and the authors have cited some of the major papers). Hence, while the new RNA linkers are potentially useful, it is not immediately obvious how much of an advantage they provide over other existing inducible methods. Overall, I would be more enthusiastic if the authors can present a stronger case for their work (e.g. by benchmarking their inventions against other inducible sgRNAs).

Major comments:

- 1) The screen appears to be hardly saturated (few constructs are found twice). Are there better designs yet to be discovered? Instead of picking 60 colonies, the authors can use Illumina sequencing to screen through thousands of colonies.
- 2) There is a need to ensure robustness in the reported results.
 - Can the authors test additional sites in the *xylA* gene (similar to *galK* - figure 1e)?
 - The authors should test sites in additional genes other than *galK* and *xylA*. All the results appear to be centered around these two genes.
- 3) There are many ways to control gene expression in *E. coli*. Figure 3f: Why don't the authors just place Cas9 under an arabinose-inducible P_{bad} promoter (or other inducible promoter)? Figure 3e: I would imagine that I can achieve a similar result by using P_{bad}-SpCas9 and Plac-SaCas9 (add arabinose first, then add IPTG later) etc - what advantage does the authors' system provide?
- 4) The authors claim that they have developed "agRNAs that overcome the issue of leakiness associated with other inducible systems." Which other inducible systems are the authors referring to (e.g. aptazyme-sgRNAs, toehold-sgRNAs etc)? Importantly, please provide the side-by-side comparison data to support the claim.
- 5) DSBs are lethal possibly because the *E. coli* cells are rapidly dividing. Can the authors' agRNAs be coupled with strategies that manipulate the cell cycle to enhance performance? (E.g. incubate cells at lower temperatures, add drug that temporarily halts the cell cycle etc.)

Minor comments:

- 1) Can the authors provide some potential explanation for why the success rate of their screen is surprisingly low?
- 2) The authors wrote that "inducible Cas9 proteins have also been developed" but only cited one paper (on arCas9). They should acknowledge other major efforts, such as split-Cas9 (PMID: 25643054), Cas9-ERT2 (PMID: 27618190), and ciCas9 (PMID: 28737741).
- 3) The authors wrote that the existing inducible Cas9 proteins "cannot support multiplexed sgRNA targeting." As written, it is unclear why this is the case because one can use an inducible SpCas9 enzyme and another inducible SaCas9 enzyme to perform multiplex editing. Are the authors trying to say that their setup only requires one Cas9 nuclease and then they can put in various inducible sgRNAs?
- 4) Briefly describe the red-white screening method (with MacConkey agar containing neutral red dye) for the uninitiated reader.

5) It would be nice to be consistent in the thresholds.

"13 out of 60 constructs from the restricted theophylline libraries showed a >50-fold increase in editing efficiency as opposed to 5 out of 150 from the initial screen."

"17 agRNA constructs showed a combined >10-fold induction and >40% editing efficiency when induced and were selected for further screening."

Why don't the authors use the same criteria in both situations?

6) Supplementary Figure 4: Why does the xylA edited chromatogram have an obviously high background?

7) The authors wrote: "The observation that several agRNAs only act at the selected galK1 site highlights that the CRISPR-Cas9 system can be selected to act at a single sequence, a feature that might be exploited for reducing off-target effects." Obviously the authors are trying to put a positive spin, but I honestly doubt if any researcher will want to work with something that only works in a very small number of cases!

8) Figure 2a-b and Figure 3e: Please provide boxplots (mean +/- standard deviation) and some statistical tests.

9) Figure 2b: Not exactly orthogonal, especially A9 (theophylline), which can be activated quite strongly by 3MX. If a user wants to use A9-theo and A9-3MX simultaneously, does this mean that the concentration of 3MX has to be lowered to avoid cross-activation? If so, it would be useful to demonstrate how to achieve better orthogonality.

10) Figure 2c-d: Can the authors provide a potential explanation for why there is a difference between their in vivo and in vitro results (e.g. in vivo, there may be transcription factors etc bound at target site, which help block Cas9 activity in absence of theo)? By the way, I think the authors are a little too humble here. The in vitro data are still informative - there is a strong difference between "no theo" and "with theo" in (d) than in (c), suggesting that the main effect is in target binding/ cleavage.

11) Figure 3a is not cited in main text.

12) Figure 3b: what are the differences in editing efficiencies between wt, immediate editing (GU19 and A9), and 1 hour delayed editing (GU19 and A9)? It is best to put this data directly next to the cfu. (In the main text, the authors wrote that ~80% editing efficiency is maintained and cited Figure 3b, but I don't see the data in that figure panel.)

13) Figure 3: the authors assessed editing efficiency 3 hr after induction. But in reality, researchers are not going to stop their experiments after 3 hours of induction. What is the optimal duration of theo treatment? Do users need to wash away theo after 3 hours? Or can they leave the inducer indefinitely in the culture media? etc

14) The authors wrote that using agRNAs, non-targeting plasmids are not enriched (page 5, line 170). This is not exactly correct because there is still some slight enrichment (81% and 63%).

15) There are scattered language errors throughout the manuscript. The authors should proofread it more carefully.

Reviewer #2:

Remarks to the Author:

The ligand-inducible gRNA functionalized with a theophylline aptamer has been used for CRISPRi.

However, this method has not been reported to be used for CRISPR-Cas genome editing due to the high leakage cleavage activity. The authors got 17 ligand-inducible gRNAs in *E. coli* against *galk* target by extensive screening with editing efficiencies over 40%. The replacement of these 17 gRNAs with N20 on *xylA* editing plasmid also showed stringent control. 3 of them (A9, GU19 and A1-gRNA) reached 40% editing efficiency against *xylA*.

By replacing the wild type gRNA targeting *galk1* with A9 or GU19-gRNA, the transformation efficiency of the CREATE plasmid increased from 10^{2-3} to 10^{7-8} CFU/ug DNA while the editing efficiency reduced to acceptable 70-80% from 90%. Moreover, the editing efficiency of CREATE plasmid mixture with 10% non-targeting gRNA dropped from 90% to 10-20% with wt gRNA but not AU or GU19-gRNA. The authors also compared several reported inducible CRISPR-Cas systems at the *galk* site, confirming that only the system developed in this manuscript was inducible while the others were actually constitutive active.

The reported C22A mutation prevented A9-gRNA from being activated by theophylline, and its 3-methylxanthine-inducible editing efficiency increased from 40% to 60%. However, the editing efficiency of GU19-gRNA(C22A) reduced to nearly zero. The authors claimed that sequential induction of GU19-gRNA targeting *xylA* and A9-gRNA(C22A) targeting *galk1* could improve the editing efficiency of both sites simultaneously, but it was not that obvious when compared with 3-methylxanthine induction alone as indicated in Fig3e and Fig S10.

The authors have made a substantial progress on the stringent inducible CRISPR-Cas genome editing system in a model microorganism. However, only 3 out of 4 targeting sites of *galk*, and *xylA* (only one target sequence was tested) were confirmed to have acceptable editing efficiency as well as the low background leakage activity with A9-gRNA. Others such as GU19 and A1-gRNA showed sequence dependence when editing *galk*. In order to confirm the usefulness of the ligand-inducible gRNA in *E. coli*, it is necessary to compare the developed system in this manuscript with other wild-type and inducible CRISPR-Cas systems on more gene targets, especially the application in the construction of multiplex CREATE libraries or multiplex CRISPR/Cas9 genome editing.

Comments:

p5, 2nd paragraph: The gene target of the CREATE plasmid with ligand-inducible gRNA should be clarified.

Fig 3e lacks wt gRNA control.

Response to Reviews.

Reviewer #1 (Remarks to the Author):

In the submitted manuscript, the authors developed novel ligand-switchable sgRNAs for genome editing in bacteria. I like the concept of a screen to identify functional theophylline-responsive or 3MX-responsive sgRNAs. However, the topic of conditional CRISPR-Cas systems is very well researched over the past few years (and the authors have cited some of the major papers). Hence, while the new RNA linkers are potentially useful, it is not immediately obvious how much of an advantage they provide over other existing inducible methods. Overall, I would be more enthusiastic if the authors can present a stronger case for their work (e.g. by benchmarking their inventions against other inducible sgRNAs).

Major comments:

1) The screen appears to be hardly saturated (few constructs are found twice). Are there better designs yet to be discovered? Instead of picking 60 colonies, the authors can use Illumina sequencing to screen through thousands of colonies.

The use of illumina sequencing is a good suggestion and something we have considered, but ultimately decided against. As reviewer #1 mentions, few constructs are found twice, which implies that there is a large number of uncharacterized, switchable agRNAs in the library. Thus, many agRNAs could be enriched in the library and detected by Illumina sequencing, each agRNA—potentially thousands--would still have to be individually cloned and characterized to determine whether it is functional, switchable and not *galK1*-specific. Therefore, given the experimental workflow of the screen, it is not easy to take full advantage of the high throughput that Illumina sequencing provides.

2) There is a need to ensure robustness in the reported results.

- Can the authors test additional sites in the *xylA* gene (similar to *galK* - figure 1e)?
- The authors should test sites in additional genes other than *galK* and *xylA*. All the results appear to be centered around these two genes.

We have focused on these genes involved in sugar metabolism because a knock-out is easily observable with the MacConkey color screen. Currently, the general consensus is that Cas9 editing efficiency is determined mainly by the spacer sequence and less by the genomic context of bacteria, which is illustrated by the fact that the sites *galK1*, *galK3* and *galK4* could be targeted by the agRNA A9, but *galK2* could not be targeted and these sites are found within the same gene but require different spacer sequences. We can certainly test more sites in the *xylA* gene or a different gene, but ultimately the result will be that some sequences can be targeted and some cannot be targeted with the agRNA, just like it is the case with the wild type gRNA/Cas9 system. Those who wish to use the agRNA will have to try a few spacers to find out which site supports efficient editing, just like researchers already do with the existing wild type gRNA/Cas9 system.

3) There are many ways to control gene expression in *E. coli*. Figure 3f: Why don't the authors just place Cas9 under an arabinose-inducible P_{bad} promoter (or other inducible promoter)? Figure 3e: I would imagine that I can achieve a similar result by using P_{bad}-SpCas9 and Plac-SaCas9 (add arabinose first, then add IPTG later) etc - what advantage does the authors' system provide?

The reference to the Supplemental Figure 1 that provides this data was missing in the manuscript and that has been corrected. In short, the pBAD promoter is too leaky to regulate Cas9 activity. From personal communication with other researchers we also know that using IPTG- or tet-inducible promoters yields the same outcome.

Page 2, Paragraph 1: ***“Due to the inherent leakiness of these systems they cannot be applied to inducible gene editing in bacteria—the same reason why inducible promoters could not be used to solve the problem^{1,19} (Supplementary Fig. 1).”***

4) The authors claim that they have developed "agRNAs that overcome the issue of leakiness associated with other inducible systems." Which other inducible systems are the authors referring to (e.g. aptazyme-sgRNAs, toehold-sgRNAs etc)? Importantly, please provide the side-by-side comparison data to support the claim.

These data are provided in Fig. 3f.

5) DSBs are lethal possibly because the E. coli cells are rapidly dividing. Can the authors' agRNAs be coupled with strategies that manipulate the cell cycle to enhance performance? (E.g. incubate cells at lower temperatures, add drug that temporarily halts the cell cycle etc.)

We grew the bacteria at 37 °C vs 30 °C but did not observe any differences in survival or editing efficiency (data not shown in manuscript). On a more theoretical note, the Cas9/gRNA complex binds very tightly to its DNA target, the main cause of dissociation from DNA is thought to be replacement of Cas9 through the helicase activity of the DNA polymerase. Dissociation of Cas9 from the DNA is also expected to be required for initiation of the homologous repair. Therefore, if DNA replication is halted, mutagenesis is also expected to be impaired.

Minor comments:

1) Can the authors provide some potential explanation for why the success rate of their screen is surprisingly low?

In Fig. 1d we show that many of the screened constructs are constitutively inactive. We suspect that this is because the selection pressure to eliminate constitutively active constructs (cell death by Cas9-mediated DSB) was much stronger than the selection pressure to eliminate constitutively inactive constructs (growth in media without metabolizable carbon source).

2) The authors wrote that "inducible Cas9 proteins have also been developed" but only cited one paper (on arCas9). They should acknowledge other major efforts, such as split-Cas9 (PMID: 25643054), Cas9-ERT2 (PMID: 27618190), and ciCas9 (PMID: 28737741).

Thanks for the additional references, they are now included in the revised manuscript.

3) The authors wrote that the existing inducible Cas9 proteins "cannot support multiplexed sgRNA targeting." As written, it is unclear why this is the case because one can use an inducible SpCas9 enzyme and another inducible SaCas9 enzyme to perform multiplex editing. Are the authors trying to say that their setup only requires one Cas9 nuclease and then they can put in various inducible sgRNAs?

We are unaware of a published regulatable SaCas9, although it is possible that one will be engineered in the future. We do agree though that it is not strictly correct to say that inducible proteins “cannot” support multiplexing and deleted the statement. The text

(page 2, paragraph 1) now reads: ***“Inducible CRISPR systems have previously been developed but are designed for CRISPRi applications and/or use in mammalian cells⁶⁻¹⁸. Due to the inherent leakiness of these systems they cannot be applied to inducible gene editing in bacteria”***

4) Briefly describe the red-white screening method (with MacConkey agar containing neutral red dye) for the uninitiated reader.

We have expanded the main text to address the screening method more clearly (page 3, paragraph 4), as follows: *“Colonies with an unedited galK site will appear red because acidification of the colony through fermentation of galactose causes the dye to turn red. Edited colonies appear white, because they cannot ferment galactose (Supplementary Fig. 2). The percentage of white colonies is considered the editing efficiency.”* Also, there is a detailed description of red-white screening in the methods section.

5) It would be nice to be consistent in the thresholds.

"13 out of 60 constructs from the restricted theophylline libraries showed a >50-fold increase in editing efficiency as opposed to 5 out of 150 from the initial screen."

"17 agRNA constructs showed a combined >10-fold induction and >40% editing efficiency when induced and were selected for further screening."

Why don't the authors use the same criteria in both situations?

Good point, this has been updated in the manuscript. To address this point, text on page 4, paragraph 1 now reads: *“Although most screened agRNA constructs were constitutively inactive (Fig. 1d), 10 out of 60 constructs from the restricted theophylline libraries showed a combined >40% editing efficiency and >10-fold increase in editing efficiency when induced (Fig. 1d) as opposed to 7 out of 150 from the initial screen.”*

6) Supplementary Figure 4: Why does the xylA edited chromatogram have an obviously high background?

We can't tell for sure, but one possible explanation might be remaining salts in the plasmid preparation that was sent for sequencing. Despite the high background, the sequence is unambiguous.

7) The authors wrote: "The observation that several agRNAs only act at the selected galK1 site highlights that the CRISPR-Cas9 system can be selected to act at a single sequence, a feature that might be exploited for reducing off-target effects." Obviously the authors are trying to put a positive spin, but I honestly doubt if any researcher will want to work with something that only works in a very small number of cases!

It is true that off-target effects of Cas9 are not a major concern in applications such as microbial strain engineering and researchers will see no reason for laboriously developing a gRNA that is site specific, we agree in that. However, in animal research or even human gene therapy the reduction of off-target effects is a major concern. If researchers in the past were willing to spend years on developing zinc-finger nucleases or TALENs for site-specific editing, they might also consider spending a few months on developing a site-specific gRNA if it reduces off-target effects encountered by wt gRNAs. We cannot predict how successful this approach would be, which is why we only hypothesize that it “might” be useful. Also, we are referring here to agRNAs like A1 that could only target a single site. agRNAs like A9 that can target multiple sites retain the flexibility associated with the CRISPR techniques.

8) Figure 2a-b and Figure 3e: Please provide boxplots (mean +/- standard deviation) and some statistical tests.

The data is provided in Supp. Fig. 5, 6 and 11 respectively.

9) Figure 2b: Not exactly orthogonal, especially A9 (theophylline), which can be activated quite strongly by 3MX. If a user wants to use A9-theo and A9-3MX simultaneously, does this mean that the concentration of 3MX has to be lowered to avoid cross-activation? If so, it would be useful to demonstrate how to achieve better orthogonality.

We concede that “orthogonal” is not technically correct, because of the cross-activation of the A9-theo agRNA, and the main text has been adjusted to reflect this point. Revised text on page 4, paragraph 3 reads: “*This not only illustrates that ligand recognition by the aptamer is necessary for agRNA activation, but also expands our toolbox by a different ligand-activatable agRNA.*” Additionally, we have modified text on page 7, paragraph 2 to read: “*Furthermore, combination of different guide sequences with aptamers that bind distinct ligands allows for multiplexing of the agRNA regulation enabling multiple, targeted mutations in a single experiment.*”

10) Figure 2c-d: Can the authors provide a potential explanation for why there is a difference between their *in vivo* and *in vitro* results (e.g. *in vivo*, there may be transcription factors etc bound at target site, which help block Cas9 activity in absence of theo)? By the way, I think the authors are a little too humble here. The *in vitro* data are still informative - there is a strong difference between “no theo” and “with theo” in (d) than in (c), suggesting that the main effect is in target binding/ cleavage.

DNA-binding proteins in the cell might definitely play a role in decreasing the leakiness of Cas9. However, it is a common problem that small molecule-binding aptamers do not transfer well between *in vivo* and *in vitro* and no definite answer has been found so far to explain this inconsistency. We hypothesize that the differences in terms of crowding and ion composition between the cellular environment and *in vitro* conditions are in large part responsible for the differences in nuclease activity. The section in the main text has been updated to clarify this, as follows: (page 5, paragraph 2) “*Although theophylline increases the nuclease activity of the RNP complex, the *in vitro* conditions greatly reduce the dynamic range of induction with theophylline. This inconsistency between functionality *in vivo* and *in vitro* might be caused by differences between the cellular environment and the conditions *in vitro*, illustrating the importance of using cell-based selections when developing cell-based tools.*”

11) Figure 3a is not cited in main text.

Thanks for pointing this out, the error has been corrected in the manuscript. Text on page 5, paragraph 3 now reads: “*We hypothesize that the stress of transformation combined with rapid generation of dsDNA synergistically leads to a high level of cell death and that temporally separating the two events can improve survival (Fig. 3a).*”

12) Figure 3b: what are the differences in editing efficiencies between wt, immediate editing (GU19 and A9), and 1 hour delayed editing (GU19 and A9)? It is best to put this data directly next to the cfu. (In the main text, the authors wrote that ~80% editing efficiency is maintained and cited Figure 3b, but I don't see the data in that figure panel.)

Thank you for pointing out this error; we have now corrected this and the panel is added in figure 3.

13) Figure 3: the authors assessed editing efficiency 3 hr after induction. But in reality, researchers are not going to stop their experiments after 3 hours of induction. What is the optimal duration of the treatment? Do users need to wash away the inducer after 3 hours? Or can they leave the inducer indefinitely in the culture media? Etc

Theophylline is non-toxic to *E. coli*, so researchers can leave it in the media as long as they want to. The main text has been updated to reflect this. Text on page 4, paragraph 3 now reads: “Even though no induction times longer than 3 hours were tested, the induction could be prolonged if desired because theophylline is non-toxic to the cells at these concentrations.”

14) The authors wrote that using agRNAs, non-targeting plasmids are not enriched (page 5, line 170). This is not exactly correct because there is still some slight enrichment (81% and 63%).

When looking at Fig. 3c, the editing efficiency of wild type sgRNAs is around 90%, but drops when transformed together with non-targeting sgRNAs. This shows that non-targeting sgRNAs are enriched in that population of cells. Because the editing efficiency of the agRNAs are the same when comparing transformation with vs. without non-targeting sgRNAs (considering the margin of error) we can infer that the non-targeting gRNAs are not enriched. This is confirmed by the sequencing results in Supp Tab. 3.

15) There are scattered language errors throughout the manuscript. The authors should proofread it more carefully.

We have made corrections throughout the manuscript that should address this issue.

Reviewer #2 (Remarks to the Author):

The ligand-inducible gRNA functionalized with a theophylline aptamer has been used for CRISPRi. However, this method has not been reported to be used for CRISPR-Cas genome editing due to the high leakage cleavage activity. The authors got 17 ligand-inducible gRNAs in *E. coli* against galK target by extensive screening with editing efficiencies over 40%. The replacement of these 17 gRNAs with N20 on xylA editing plasmid also showed stringent control. 3 of them (A9, GU19 and A1-gRNA) reached 40% editing efficiency against xylA.

By replacing the wild type gRNA targeting galK1 with A9 or GU19-gRNA, the transformation efficiency of the CREATE plasmid increased from 10^{2-3} to 10^{7-8} CFU/ug DNA while the editing efficiency reduced to acceptable 70-80% from 90%. Moreover, the editing efficiency of CREATE plasmid mixture with 10% non-targeting gRNA dropped from 90% to 10-20% with wt gRNA but not AU or GU19-gRNA. The authors also compared several reported inducible CRISPR-Cas systems at the galK site, confirming that only the system developed in this manuscript was inducible while the others were actually constitutive active.

The reported C22A mutation prevented A9-gRNA from being activated by theophylline, and its 3-methylxanthine-inducible editing efficiency increased from 40% to 60%. However, the editing efficiency of GU19-gRNA(C22A) reduced to nearly zero. The authors claimed that sequential

induction of GU19-gRNA targeting xylA and A9-gRNA(C22A) targeting galK1 could improve the editing efficiency of both sites simultaneously, but it was not that obvious when compared with 3-methylxanthine induction alone as indicated in Fig3e and Fig S10.

It is true that using only 3-methylxanthine is surprisingly effective in generating combinatorial mutations and we do not have a clear explanation as to why this is the case. We would like to highlight that expressing both agRNAs from one plasmid is still necessary for generating the combinatorial mutations as targeting more than one site with the wild type sgRNAs leads to zero survival of the cells.

The authors have made a substantial progress on the stringent inducible CRISPR-Cas genome editing system in a model microorganism. However, only 3 out of 4 targeting sites of galK, and xylA (only one target sequence was tested) were confirmed to have acceptable editing efficiency as well as the low background leakage activity with A9-gRNA. Others such as GU19 and A1-gRNA showed sequence dependence when editing galK. In order to confirm the usefulness of the ligand-inducible gRNA in *E. coli*, it is necessary to compare the developed system in this manuscript with other wild-type and inducible CRISPR-Cas systems on more gene targets, especially the application in the construction of multiplex CREATE libraries or multiplex CRISPR/Cas9 genome editing.

While it is true that the agRNAs cannot target all sites, the same is true for wild type sgRNAs. In either case, the researchers who ultimately use this technology will have to target a few sites to find a site that permits high-efficiency editing. We started out with sites that were known to allow high-efficiency editing with the wild type sgRNA and the agRNAs apparently have different spacer sequence requirements, but that does not change how a researcher would have to test target sites when preparing an experiment. Also, testing more target sites would not solve this intrinsic issue.

We agree that it would be interesting to use the agRNAs for a genome editing library. However, work from other groups shows that the editing efficiencies drop to 0-10% when CRISPR is used for library-scale genome editing in bacteria without any selection. We multiplexed 4 sites on the galK gene and deep sequenced the gene after editing, which gave a similar, low efficiency result. The data is now shown in Supp. Fig. 10 and a paragraph was added to address this issue:

Page 6, paragraph 2: “To ascertain whether agRNAs can facilitate creation of libraries of variants, the percentage of non-functional sgRNAs in the plasmid mix was varied from 0-50% in combination with 4 different plasmids, each expressing an agRNA targeting a different site on the galK gene. After induction of editing, cells were plated and the edits in the galK gene were analyzed via next-generation sequencing. In the absence of non-functional sgRNAs, editing efficiencies with the agRNAs were as low as with the sgRNAs, around 1-10% (Supplementary Fig. 10). Increasing amounts of non-functional sgRNAs further reduced the editing efficiencies. We cannot explain the drastic loss of editing efficiency in light of the single agRNA experiment (vide supra). However, we suspect that cells might be transformed with more than one plasmid such that the sgRNAs would target more than one site. This would lead to multiple genomic DNA breaks which places a large burden on the cells so that any potential “escapees” can take over the bacterial population. E. coli does not typically survive two simultaneous dsDNA breaks, even with induction of the λ-red proteins to facilitate homologous repair¹.”

Comments:

p5, 2nd paragraph: The gene target of the CREATE plasmid with ligand-inducible gRNA should be clarified.

Thank you for this comment and we have clarified as follows (Page 5, paragraph 3):
“Targeting the galK1 site with the A9 or GU19 agRNAs resulted in a remarkable 10⁴-fold increase in number of transformants while maintaining ~80% editing efficiencies (Fig. 3b).”

Fig 3e lacks wt gRNA control.

When two sites are targeted by Cas9 with wt sgRNA, no cells survive the gene editing procedure, therefore it was impossible to obtain editing efficiency values with the wt sgRNA for these experiments. We included an additional sentence on this issue in the main text:

Page 7, paragraph 1: *“When targeting both genes with sgRNAs, no transformants were observed because E. coli rarely survives two simultaneous dsDNA breaks. Therefore, no editing efficiencies could be calculated for the sgRNAs.”*

Reviewers' Comments:

Reviewer #1:

Remarks to the Author:

The authors submitted a revised manuscript describing their work on the development of inducible genome editing systems for bacteria using theophylline-inducible and 3MX-inducible aptamers. Overall, the revised manuscript is much improved and the authors have addressed many of my previous comments satisfactorily.

I only have one remaining concern, which has been pointed out earlier. Essentially all the results in the manuscript center around the *galK* and *xylA* genes. Yes, I understand that it's convenient to use *galK* and *xylA* for the purpose of technology development. But at the end, a reader will be asking himself/herself: "Should I try this method myself?" I would strongly encourage the authors to make an effort to target some non-essential genes that will result in an obvious phenotype. For example, the authors can target chemotaxis genes (resulting in loss of motility) or some cell division genes (resulting in concatenated cells or loss of cell shape). If the agRNAs (e.g. A9 or GU19) work well, these demonstrations should be straightforward to carry out and can be completed in a month or so.

Minor comments:

1) I suggest shortening this header "Screening of constructs from the selection yields switchable agRNAs that can be protospacer-dependent or -independent" to "Screening of constructs from the selection yields switchable agRNAs"

2) In the paragraph starting with "The 17 promising agRNA constructs were selected for further analysis", I would put in a sentence stating that all 17 agRNAs showed minimal activity at the *galK* and *xylA* loci in the absence of theophylline (Supplementary Figure 3).

3) The authors wrote that "theophylline is non-toxic to the cells at these concentrations." It would be good to provide a citation for this claim.

4) The authors wrote "A1, A9 and GU19 agRNAs all exhibit the same apparent binding affinity to the Cas9 protein as the wild type sgRNA" - I suggest replacing the word "same" with "similar". The binding affinities are not all exactly the same.

5) Figure 2c/ Supplementary Figure 7: For completeness sake, I suggest showing the best fit lines for A1 and GU19 agRNAs as well (besides A9 agRNA).

6) The authors wrote "Although theophylline increases the nuclease activity of the RNP complex, the in vitro conditions greatly reduce the dynamic range of induction with theophylline." - I suggest re-writing as "Indeed, theophylline did clearly increase the nuclease activity of the RNP complex. Nevertheless, the in vitro conditions greatly reduce the dynamic range of induction with theophylline."

Reviewer #2:

Remarks to the Author:

As I said in the first round of review: "the authors have made a useful attempt in the inducible CRISPR-Cas genome editing system in the microbial model organism, *E. coli*. However, only the 3 out of 4 targeting sites of *galK*, and *xylA* (only one target sequence was tested) were confirmed to have acceptable editing efficiency as well as the low background leakage activity with A9-gRNA. Others such as GU19 and A1-gRNA showed sequence dependence when editing *galK*. In order to confirm the general practicality of the ligand-inducible gRNA in *E. coli*, "

1)"it is necessary to compare the developed system in this manuscript with other wild-type and inducible CRISPR-Cas systems on more gene targets, ... "

2)"...especially the application in the construction of multiplex CREATE libraries or ..."

3)"...multiplex CRISPR/Cas9 genome editing."

In the resubmitted manuscript,

1)Authors said: " While it is true that the agRNAs cannot target all sites, the same is true for wild type sgRNAs. In either case, the researchers who ultimately use this technology will have to target a few sites to find a site that permits high-efficiency editing. We started out with sites that were known to allow high-efficiency editing with the wild type sgRNA and the agRNAs apparently have different spacer sequence requirements, but that does not change how a researcher would have to test target sites when preparing an experiment. Also, testing more target sites would not solve this intrinsic issue." That is to say, agRNA must be screened for each target with shapes for efficient editing and low leakage activity, which will result in unpredictable workload.. I believe few people would like to do so unless such an agRNA is essential in certain scenarios, such as 2) CREATE or 3) multiplex CRISPR/Cas9 genome editing. However,

2) since each agRNA must be selected from a small or big library, the use of agRNA in CREATE does not appear to be feasible.. Furthermore, even with pre-screened low-leakage and high-editing active agRNA sets, only four agRNA combinations resulted in editing efficiencies as low as 10% (Fig 10S). Not to mention that building a library requires a collection of at least thousands of agRNAs.

3) The multiplex editing efficiency of the four agRNA sets is less than 10%. In contrast, the conventional CRISPR-Cas methods achieved 47% multigene editing efficiency against 3 targets simultaneously (Jiang, Y., et al. (2015). "Multigene editing in the Escherichia coli Genome using the CRISPR-Cas9 system." *Appl Environ Microbiol.* 81, 7, 2506), not to mention the efficiency of simultaneous editing of 6 targets reached 7/8 with baseditor (Fig 4b, Banno, S., Nishida, K., Arazoe, T., Mitsunobu, H. , and Kondo, A. (2018) Deaminase-mediated multiplex genome editing in Escherichia coli, *Nature Microbiology* 3, 423-429). Therefore, agRNA is not competent for multiplex genome editing at its current performance.

Taken together, I did not see substantial improvements in this resubmitted version.

Reviewer #1 (Remarks to the Author):

The authors submitted a revised manuscript describing their work on the development of inducible genome editing systems for bacteria using theophylline-inducible and 3MX-inducible aptamers. Overall, the revised manuscript is much improved and the authors have addressed many of my previous comments satisfactorily.

I only have one remaining concern, which has been pointed out earlier. Essentially all the results in the manuscript center around the *galK* and *xylA* genes. Yes, I understand that it's convenient to use *galK* and *xylA* for the purpose of technology development. But at the end, a reader will be asking himself/herself: "Should I try this method myself?" I would strongly encourage the authors to make an effort to target some non-essential genes that will result in an obvious phenotype. For example, the authors can target chemotaxis genes (resulting in loss of motility) or some cell division genes (resulting in concatenated cells or loss of cell shape). If the agRNAs (e.g. A9 or GU19) work well, these demonstrations should be straightforward to carry out and can be completed in a month or so.

We strongly agree with Reviewer #1 in that targeting other genes to produce different phenotypes would add to the comprehensiveness of the manuscript. However, considering the usual expectations around studies that are centered on synthetic biology and the design of new CRISPR tools, we believe that additional experiments are not required for the completeness of the manuscript. Out of comparable studies, only a single one (Liu, Y. *et al.* Directing cellular information flow via CRISPR signal conductors. *Nat. Methods* 13, 938–944 (2016)) presents a strong biological application and phenotype. All other studies were restricted to targeting reporter genes such as fluorescent proteins, luciferase, *lacZ*, etc. and/or measuring the targeting of endogenous genes by sequencing or RT-PCR. To illustrate this point, we compiled a list of these studies that are also referenced in the manuscript:

Study	Targeted reporter genes	Targeted endogenous genes	Showed biological application/Produced new phenotype
Liu, Y. et al. Nat. Methods , 2016	Yes	Yes	Yes
Oakes, B. L. et al. , Nat Biotechnol. , 2016	Yes	No	No
Kundert, K. et al. , Nat. Commun. , 2019	Yes	No	No
Ferry, Q. R. V. et al. , Nat. Commun. , 2017	Yes	Yes	No
Tang, W. et al. , Nat. Commun. 2017	Yes	Yes	No
Rose, J. C. et al. , Nat. Methods , 2017	Yes	Yes	No
Liu, K. I. et al. , Nat. Chem. Biol. , 2017	Yes	Yes	No
Zetsche, B. et al. , Nat. Biotechnol. , 2015	No	Yes	No
Nihongaki, Y. et al. , Nat Biotechnol. , 2015	No	Yes	No
Hemphill, J. et al. , JACS , 2015	Yes	Yes	No
Richter, F. et al. , NAR , 2016	Yes	No	No
Maji, B. et al. , Nat. Chem. Biol. , 2017	Yes	Yes	No
Siu, K.-H. et al. , Nat. Chem. Biol. , 2018	Yes	No	No

Minor comments:

We agree with all the comments below and have incorporated the suggested changes. The comments were helpful and greatly appreciated. Only comment 5) could not be addressed, as explained below.

1) I suggest shortening this header "Screening of constructs from the selection yields switchable agRNAs that can be protospacer-dependent or -independent" to "Screening of constructs from the selection yields switchable agRNAs"

We have shortened the header.

2) In the paragraph starting with "The 17 promising agRNA constructs were selected for further analysis", I would put in a sentence stating that all 17 agRNAs showed minimal activity at the *galK* and *xylA* loci in the absence of theophylline (Supplementary Figure 3).

We made the following changes:

Page 4, Paragraph 2: “All 17 agRNAs showed low editing activity of the *galK* and *xylA* genes in the absence of ligand, but only two agRNAs (A9 and GU19) showed an editing efficiency at *xylA* similar to the *galK1* site [...]”

3) The authors wrote that "theophylline is non-toxic to the cells at these concentrations." It would be good to provide a citation for this claim.

We have added a citation (Berens, C., Groher, F. & Suess, B. RNA aptamers as genetic control devices: The potential of riboswitches as synthetic elements for regulating gene expression. *Biotechnol. J.* 10, 246–257 (2015)).

4) The authors wrote "A1, A9 and GU19 agRNAs all exhibit the same apparent binding affinity to the Cas9 protein as the wild type sgRNA" - I suggest replacing the word "same" with "similar". The binding affinities are not all exactly the same.

The words have been changed:

Page 5, Paragraph 2: “all exhibit similar apparent binding affinities”

5) Figure 2c/ Supplementary Figure 7: For completeness sake, I suggest showing the best fit lines for A1 and GU19 agRNAs as well (besides A9 agRNA).

While we certainly see the point of showing the other fit lines in the graph, they create an incomprehensible visual mess if overlaid in the same panel. Therefore, we ultimately decided it would be best to just include a table as the raw data for the binding assays is submitted in a separate file with the manuscript in any case.

6) The authors wrote "Although theophylline increases the nuclease activity of the RNP complex, the in vitro conditions greatly reduce the dynamic range of induction with theophylline." - I suggest re-writing as "Indeed, theophylline did clearly increase the nuclease activity of the RNP complex. Nevertheless, the in vitro conditions greatly reduce the dynamic range of induction with theophylline."

The wording was changed to:

Page 5, Paragraph 2: “*Theophylline did increase the nuclease activity of the RNP complex, however, the in vitro conditions greatly reduced the dynamic range of induction with theophylline.*”

Reviewer #2 (Remarks to the Author):

As I said in the first round of review: “the authors have made a useful attempt in the inducible CRISPR-Cas genome editing system in the microbial model organism, *E. coli*. However, only the 3 out of 4 targeting sites of *galK*, and *xylA* (only one target sequence was tested) were confirmed to have acceptable editing efficiency as well as the low background leakage activity with A9-gRNA. Others such as GU19 and A1-gRNA showed sequence dependence when editing *galK*. In order to confirm the general practicality of the ligand-inducible gRNA in *E. coli*, ”
1)“it is necessary to compare the developed system in this manuscript with other wild-type and inducible CRISPR-Cas systems on more gene targets, ... ”

2)“...especially the application in the construction of multiplex CREATE libraries or ...”

3)“...multiplex CRISPR/Cas9 genome editing.”

In the resubmitted manuscript,

1)Authors said: ” While it is true that the agRNAs cannot target all sites, the same is true for wild type sgRNAs. In either case, the researchers who ultimately use this technology will have to target a few sites to find a site that permits high-efficiency editing. We started out with sites that were known to allow high-efficiency editing with the wild type sgRNA and the agRNAs apparently have different spacer sequence requirements, but that does not change how a researcher would have to test target sites when preparing an experiment. Also, testing more target sites would not solve this intrinsic issue.” That is to say, agRNA must be screened for each target with shapes for efficient editing and low leakage activity, which will result in unpredictable workload.. I believe few people would like to do so unless such an agRNA is essential in certain scenarios, such as 2) CREATE or 3) multiplex CRISPR/Cas9 genome editing. However,

Reviewer #2 expresses the concern that screening of target sites might create an unpredictable workload that few people would like to take on. We agree that ease and usability is a major concern when developing new research tools. However, as we pointed out in the passage cited by the reviewer, the workload associated with agRNA is comparable to the one associated with the typical sgRNAs or even siRNAs. In all three cases, different target sites should be tested to ensure high efficiency and this has not dissuaded researchers from using sgRNAs or siRNAs in the past.

2) since each agRNA must be selected from a small or big library, the use of agRNA in CREATE does not appear to be feasible.. Furthermore, even with pre-screened low-leakage and high-editing active agRNA sets, only four agRNA combinations resulted in editing efficiencies as low as 10% (Fig 10S). Not to mention that building a library requires a collection of at least thousands of agRNAs.

The process of selecting switchable agRNAs is only supposed to be part of this study and not part of the workflow for other researchers. We expect other researchers to be able to utilize our successful constructs such as the A9 agRNA and simply retarget them like any other CRISPR gRNA.

3) The multiplex editing efficiency of the four agRNA sets is less than 10%. In contrast, the conventional CRISPR-Cas methods achieved 47% multigene editing efficiency against 3 targets simultaneously (Jiang, Y., et al. (2015). "Multigene editing in the Escherichia coli Genome using the CRISPR-Cas9 system." *Appl Environ Microbiol.* 81, 7, 2506), not to mention the efficiency of simultaneous editing of 6 targets reached 7/8 with baseditor (Fig 4b, Banno, S., Nishida, K., Arazoe, T., Mitsunobu, H. , and Kondo, A. (2018) Deaminase-mediated multiplex genome editing in Escherichia coli, *Nature Microbiology* 3, 423-429). Therefore, agRNA is not competent for multiplex genome editing at its current performance.

We agree that agRNAs combined with the CREATE platform cannot currently be used to achieve high editing efficiencies in a multiplexed manner. However, this has been reported to be an intrinsic issue related to multiplex-editing platforms such as CREATE or MAGE that target not two or three but hundreds of sites. Achieving high editing efficiencies for such multiplexed gene editing is a long-standing goal and we believe that neither we nor the studies cited above have achieved that goal. Changing DNA sequences with base editors is based on a fundamentally different mechanism that brings its own advantages and disadvantages, so we do not think these studies are directly comparable. Jiang et al. achieved high efficiencies on three targets by providing three donor templates in one plasmid, so this approach is more comparable to our experiments graphed in Fig. 3e but not the deep-seq experiments. Jiang et al. also state in their manuscript that “*We did not attempt multiple gene deletions or insertions of more than three genes because the cloning procedure for pTargetT was complicated and time-consuming when multiple donor DNAs were included. The method will not have the level of efficiency needed for metabolic engineering of an industrially relevant strain.*” They attempt to solve this issue with a different method but concede that “*We attempted to perform double-gene editing (Table 2, experiment 11) by combined deletion of locus kefB with the insertion of fevgAS into locus yjcS, but we obtained no double mutation.*” The fact that agRNAs did not solve this issue does not diminish their capacity to regulate gene editing with unprecedented stringency as shown in Fig. 3f.

Taken together, I did not see substantial improvements in this resubmitted version.

While we respect the opinion of the reviewer, the authors believe that significant improvements to the manuscript have been made during two rounds of review.

Reviewer #1 (Remarks to the Author):

The authors submitted a revised manuscript describing their work on the development of inducible genome editing systems for bacteria using theophylline-inducible and 3MX-inducible aptamers. Overall, the revised manuscript is much improved and the authors have addressed many of my previous comments satisfactorily.

I only have one remaining concern, which has been pointed out earlier. Essentially all the results in the manuscript center around the *galK* and *xylA* genes. Yes, I understand that it's convenient to use *galK* and *xylA* for the purpose of technology development. But at the end, a reader will be asking himself/herself: "Should I try this method myself?" I would strongly encourage the authors to make an effort to target some non-essential genes that will result in an obvious phenotype. For example, the authors can target chemotaxis genes (resulting in loss of motility) or some cell division genes (resulting in concatenated cells or loss of cell shape). If the agRNAs (e.g. A9 or GU19) work well, these demonstrations should be straightforward to carry out and can be completed in a month or so.

We strongly agree with Reviewer #1 in that targeting other genes to produce different phenotypes would add to the comprehensiveness of the manuscript. We should note that within comparable studies, only a single one (Liu, Y. *et al.* Directing cellular information flow via CRISPR signal conductors. *Nat. Methods* 13, 938–944 (2016)) presents a strong biological application and phenotype. All other studies were restricted to targeting reporter genes such as fluorescent proteins, luciferase, *lacZ*, etc. and/or measuring the targeting of endogenous genes by sequencing or RT-PCR. To illustrate this point, we compiled a list of these studies that are also referenced in the manuscript:

Study	Targeted reporter genes	Targeted endogenous genes	Showed biological application/Produced new phenotype
Liu, Y. et al. Nat. Methods , 2016	Yes	Yes	Yes
Oakes, B. L. et al. , Nat Biotechnol. , 2016	Yes	No	No
Kundert, K. et al. , Nat. Commun. , 2019	Yes	No	No
Ferry, Q. R. V. et al. , Nat. Commun. , 2017	Yes	Yes	No
Tang, W. et al. , Nat. Commun. 2017	Yes	Yes	No
Rose, J. C. et al. , Nat. Methods , 2017	Yes	Yes	No
Liu, K. I. et al. , Nat. Chem. Biol. , 2017	Yes	Yes	No
Zetsche, B. et al. , Nat. Biotechnol. , 2015	No	Yes	No
Nihongaki, Y. et al. , Nat Biotechnol. , 2015	No	Yes	No
Hemphill, J. et al. , JACS , 2015	Yes	Yes	No
Richter, F. et al. , NAR , 2016	Yes	No	No
Maji, B. et al. , Nat. Chem. Biol. , 2017	Yes	Yes	No
Siu, K.-H. et al. , Nat. Chem. Biol. , 2018	Yes	No	No

However, to fully address the reviewer's concerns, we additionally targeted the genes *ispC* and *lacZ* with the agRNA A9. *ispC* has been edited in *E. coli* to investigate resistance against the malarial drug fosmidomycin (Pines *et al.*, *ACS SynBio* 2018) and although resistance to fosmidomycin could have

been used as a screen for editing, we found Sanger sequencing to be more reliable. Editing of *lacZ* was easily quantifiable with an X-gal screen and although the editing efficiencies were relatively low, the same was observed for editing with wild type sgRNAs. A collective overview of editing efficiencies with agRNA A9 can be seen below (and is presented as Figure 1e in the revised manuscript). Out of the nine genomic sites targeted, only two (*galk2* and *lacZ2*) could not be edited. At the seven sites that could be edited, the agRNA was consistently activated by theophylline. This indicates that the agRNA can be easily targeted to new sites across the genome.

Minor comments:

We agree with all the comments below and have incorporated the suggested changes. The comments were helpful and greatly appreciated. Only comment 5) could not be incorporated as explained below.

1) I suggest shortening this header "Screening of constructs from the selection yields switchable agRNAs that can be protospacer-dependent or -independent" to "Screening of constructs from the selection yields switchable agRNAs"

We have shortened the header.

2) In the paragraph starting with "The 17 promising agRNA constructs were selected for further analysis", I would put in a sentence stating that all 17 agRNAs showed minimal activity at the *galk* and *xylA* loci in the absence of theophylline (Supplementary Figure 3).

We made the following changes:

Page 4, Paragraph 2: "All 17 agRNAs showed low editing activity of the *galk* and *xylA* genes in the absence of ligand, but only two agRNAs (A9 and GU19) showed an editing efficiency at *xylA* similar to the *galk1* site [...]"

3) The authors wrote that "theophylline is non-toxic to the cells at these concentrations." It would be good to provide a citation for this claim.

We have added a citation (Berens, C., Groher, F. & Suess, B. RNA aptamers as genetic control devices: The potential of riboswitches as synthetic elements for regulating gene expression. *Biotechnol. J.* 10, 246–257 (2015)).

4) The authors wrote "A1, A9 and GU19 agRNAs all exhibit the same apparent binding affinity to the Cas9 protein as the wild type sgRNA" - I suggest replacing the word "same" with "similar". The binding affinities are not all exactly the same.

The words have been changed:

Page 5, Paragraph 2: *"all exhibit similar apparent binding affinities"*

5) Figure 2c/ Supplementary Figure 7: For completeness sake, I suggest showing the best fit lines for A1 and GU19 agRNAs as well (besides A9 agRNA).

While we certainly see the point of showing the other fit lines in the graph, they create an incomprehensible visual mess if overlaid in the same panel. If all 5 lines +/- ligand are displayed in different panels, it blows up the figure disproportionately with 5 graphs that all look identical and thus redundant. Therefore, we ultimately decided it would be best to just include a table as the raw data for the binding assays is submitted in a separate file with the manuscript in any case.

6) The authors wrote "Although theophylline increases the nuclease activity of the RNP complex, the in vitro conditions greatly reduce the dynamic range of induction with theophylline." - I suggest re-writing as "Indeed, theophylline did clearly increase the nuclease activity of the RNP complex. Nevertheless, the in vitro conditions greatly reduce the dynamic range of induction with theophylline."

The wording was changed to:

Page 5, Paragraph 2: *"Theophylline did increase the nuclease activity of the RNP complex, however, the in vitro conditions greatly reduced the dynamic range of induction with theophylline."*

Reviewer #2 (Remarks to Authors)

As I said in the first round of review: "the authors have made a useful attempt in the inducible CRISPR-Cas genome editing system in the microbial model organism, *E. coli*. However, only the 3 out of 4 targeting sites of *galk*, and *xylA* (only one target sequence was tested) were confirmed to have acceptable editing efficiency as well as the low background leakage activity with A9-gRNA. Others such as GU19 and A1-gRNA showed sequence dependence when editing *galk*. In order to confirm the general practicality of the ligand-inducible gRNA in *E. coli*,"

1)"it is necessary to compare the developed system in this manuscript with other wild-type and inducible CRISPR-Cas systems on more gene targets, ..."

2)“...especially the application in the construction of multiplex CREATE libraries or ...”

3)“...multiplex CRISPR/Cas9 genome editing.”

In the resubmitted manuscript,

1)Authors said: ” While it is true that the agRNAs cannot target all sites, the same is true for wild type sgRNAs. In either case, the researchers who ultimately use this technology will have to target a few sites to find a site that permits high-efficiency editing. We started out with sites that were known to allow high-efficiency editing with the wild type sgRNA and the agRNAs apparently have different spacer sequence requirements, but that does not change how a researcher would have to test target sites when preparing an experiment. Also, testing more target sites would not solve this intrinsic issue.” That is to say, agRNA must be screened for each target with shapes for efficient editing and low leakage activity, which will result in unpredictable workload.. I believe few people would like to do so unless such an agRNA is essential in certain scenarios, such as 2) CREATE or 3) multiplex CRISPR/Cas9 genome editing. However,

Reviewer #2 expresses the concern that screening of target sites might create an unpredictable workload that few people would like to take on. We agree that ease and usability is a major concern when developing new research tools. However, as we pointed out in the passage cited by the reviewer, the workload associated with agRNA is comparable to the one associated with the typical sgRNAs or even siRNAs. In all three cases, different target sites should be tested to ensure high efficiency and this has not dissuaded researchers from using sgRNAs or siRNAs in the past.

2) since each agRNA must be selected from a small or big library, the use of agRNA in CREATE does not appear to be feasible.. Furthermore, even with pre-screened low-leakage and high-editing active agRNA sets, only four agRNA combinations resulted in editing efficiencies as low as 10% (Fig 10S). Not to mention that building a library requires a collection of at least thousands of agRNAs.

The process of selecting switchable agRNAs is only supposed to be part of this study and not part of the workflow for other researchers. We expect other researchers to be able to utilize our successful constructs such as the A9 agRNA and simply retarget them like any other CRISPR gRNA.

3) The multiplex editing efficiency of the four agRNA sets is less than 10%. In contrast, the conventional CRISPR-Cas methods achieved 47% multigene editing efficiency against 3 targets simultaneously (Jiang, Y., et al. (2015). "Multigene editing in the Escherichia coli Genome using the CRISPR-Cas9 system." Appl Environ Microbiol. 81, 7, 2506), not to mention the efficiency of simultaneous editing of 6 targets reached 7/8 with baseditor (Fig 4b, Banno, S., Nishida, K., Arazoe, T., Mitsunobu, H. , and Kondo, A. (2018) Deaminase-mediated multiplex genome editing in Escherichia coli, Nature Microbiology 3, 423-429). Therefore, agRNA is not competent for multiplex genome editing at its current performance.

We agree that agRNAs combined with the CREATE platform cannot currently be used to achieve high editing efficiencies in a multiplexed manner. However, this has been reported to be an intrinsic issue related to multiplex-editing platforms such as CREATE or MAGE that target not two or three but hundreds of sites. Achieving high editing efficiencies for such multiplexed gene editing is a long-standing goal and we believe that neither we nor the studies cited above have achieved that goal. Changing DNA sequences with base editors is based on a fundamentally mechanism that brings its own advantages and disadvantages, so we do not think these studies are directly comparable. Jiang et al. achieved high efficiencies on three targets by providing three donor templates in one plasmid, so this approach is more comparable to our experiments graphed in Fig. 3e but not to the deep-seq experiments. Jiang et al. also state in their manuscript that *“We did not attempt multiple gene deletions or insertions of more than three genes because the cloning procedure for pTargetT was complicated and time-consuming when multiple donor DNAs were included. The method will not have the level of efficiency needed for metabolic engineering of an industrially relevant strain.”* They attempt to solve this issue with a different method but concede that *“We attempted to perform double-gene editing (Table 2, experiment 11) by combined deletion of locus kefB with the insertion of evgAS into locus yjcS, but we obtained no double mutation.”* The fact that agRNAs did not solve this issue does not diminish their capacity to regulate gene editing with unprecedented stringency as shown in Fig. 3f.

Taken together, I did not see substantial improvements in this resubmitted version.

REVIEWERS' COMMENTS:

Reviewer #1 (Remarks to the Author):

The authors submitted a second revised manuscript. The main improvement here is the testing of their agRNAs at additional sites in lacZ and ispC. Overall, I am satisfied with the new results. As with all new technology, the agRNA approach comes with its own strengths and weaknesses. It will be good to put the technology out in the public domain and let others try it out.

Just a quick note: the additional text on page 4 (in red) is rather unwieldy. The authors can simply write "Two of the three lacZ-targeting agRNAs exhibited similar editing efficiencies to unmodified sgRNAs, while the agRNA targeting ispC was ~80% as efficient as the corresponding sgRNA."

- Meng How Tan

Reviewer #2 (Remarks to the Author):

In their rebuttal letter, the authors agreed that agRNA is not competent for either CREATE or multiplex genome editing. So, what else can agRNA be used for? Currently, I can't figure out any scenario in which agRNA should show significant advantages to other methods in this resubmitted version.

Our responses to the reviewers are detailed below, with our response highlighted in bold.

Reviewer #2 commented: *"Just a quick note: the additional text on page 4 (in red) is rather unwieldy. The authors can simply write "Two of the three lacZ-targeting agRNAs exhibited similar editing efficiencies to unmodified sgRNAs, while the agRNA targeting ispC was ~80% as efficient as the corresponding sgRNA."*

We certainly see the point made and changed the text to:

Two of the lacZ-targeting agRNAs and the ispC-targeting agRNA exhibited editing efficiencies similar to unchanged sgRNAs. In total, out [...]